# High mitogenic stimulation arrests angiogenesis

Samuel Pontes-Quero [1], Macarena Fernández-Chacón [1], Wen Luo[1], Federica Francesca Lunella[1], Verónica Casquero-Garcia [1], Irene Garcia-Gonzalez [1], Ana Hermoso[1], Susana F. Rocha[1], Mayank Bansal [1] & Rui Benedito[1]

Appropriate therapeutic modulation of endothelial proliferation and sprouting is essential for the effective inhibition of angiogenesis in cancer or its induction in cardiovascular disease. The current view is that an increase in growth factor concentration, and the resulting mitogenic activity, increases both endothelial proliferation and sprouting. Here, we modulate mitogenic stimuli in different vascular contexts by interfering with the function of the VEGF and Notch signalling pathways at high spatiotemporal resolution in vivo. Contrary to the prevailing view, our results indicate that high mitogenic stimulation induced by VEGF, or Notch inhibition, arrests the proliferation of angiogenic vessels. This is due to the existence of a bell-shaped dose-response to VEGF and MAPK activity that is counteracted by Notch and p21, determining whether endothelial cells sprout, proliferate, or become quiescent. The identified mechanism should be considered to achieve optimal therapeutic modulation of angiogenesis.

[1] Molecular Genetics of Angiogenesis Group, Centro Nacional de Investigaciones Cardiovasculares (CNIC), Madrid 28029, Spain. Correspondence and requests for materials should be addressed to R.B. (email: Rui.benedito@cnic.es)

Blood vessel formation requires tight coordination of molecular and cellular processes to ensure that new endothelial cells (ECs) are generated at the right time, pace and place. Numerous studies conducted in embryos, organs, and tumours have established the importance for angiogenesis of the VEGF and Dll4/Notch signalling pathways[1–8].

Growing or hypoxic tissues secrete vascular endothelial growth factor (VEGF). Activation of VEGFR2, the most important VEGF receptor, then triggers a series of phosphorylation cascades, including ERK (MAPK) activation, that are considered essential for EC motility and proliferation. The ECs at the tip of the vascular front, and closest to the source of VEGF, express higher levels of several genes, including Esm1, Angpt2 and the Notch ligand Dll4[9,10], giving them a distinct molecular signature from that of adjacent stalk cells. Notch activity is thought to be relatively higher in stalk cells and inhibits endothelial sprouting and proliferation, and is therefore assumed to be a negative regulator of VEGF function in these cells. Tip-cell specification and EC proliferation are two related processes considered to be positively regulated by VEGF and negatively regulated by Notch. According to this model, tip cells should proliferate more than stalk cells because they have higher-VEGF signalling and lower-Notch signalling. However, in vivo studies suggested that tip cells do not proliferate, or proliferate much less than stalk cells[11,12]. In contrast to these findings, live imaging during zebrafish intersegmental vessels development showed that in some contexts tip cells can divide equally as well as stalk cells, while migrating and sprouting[4]. The inter-regulation of EC sprouting and proliferation dynamics by VEGF and Notch has been difficult to adjust to a simple model in which actively sprouting ECs do not proliferate, since these cells receive the highest VEGF stimulus and are thought to have low-Notch signalling, two conditions believed to be pro-mitogenic.

Here, we use inducible fluorescent genetic mosaic (ifgMosaic) mouse lines[13] and different pharmacological treatments to define at high cellular and temporal resolution the role of the Notch and VEGF signalling pathways in the proliferative behaviour of ECs. This approach reveals that ECs have a bell-shaped dose–response to mitogenic stimuli in vivo that is highly dependent on the vascular developmental context. This is due to a cell-cycle checkpoint molecular mechanism that is elicited when VEGF stimulation is high or Notch signalling is low during angiogenesis. In these conditions, ECs have high-ERK activity, which induces the expression of the cell-cycle inhibitor p21, and the ensuing cell-cycle arrest enhances endothelial sprouting and migration, but ultimately blocks angiogenesis due to the inhibition of endothelial proliferation. Our study revisits the important role of Notch and VEGF in the balance of endothelial proliferation and sprouting during angiogenesis, and provides significantly more molecular, cellular and temporal resolution on the underlying mechanism.

## Results

### Angiogenic ECs proliferate less after Notch loss-of-function.
Studies with pharmacological compounds or with classical knockout or conditional mouse lines revealed that loss-of-function (LOF) of Dll4, Notch1 or Rbpj results in severe vascular developmental defects and a marked delay in tumour growth due to an abnormal increase in angiogenesis[5,6,11,14–20]. Inhibition of Notch signalling in human venous ECs (HUVECs) or mouse EC lines increases their proliferation[21,22] (see also Supplementary Fig. 1a, b). However, there is less consensus on the effect of Notch inhibition on EC proliferation in vivo. Some groups reported no significant differences in the number of ECs in s-phase (BrdU+) in retinas of Dll4 heterozygous mice or after treatment with a general γ-secretase inhibitor (DAPT)[11,20],

whereas others have seen an increase in the frequency of BrdU+ or Ki67 + ECs in retina vessels of mice treated with different Notch signalling inhibitors (γ-secretase inhibitor or Dll4-Fc proteins)[5,22,23]. Live imaging of intersegmental arteries development showed an increase in the number of ECs in zebrafish embryos with a morpholino-induced reduction of Dll4 and Rbpj expression[4].

Rbpj is the main transcription factor that associates with all four Notch intracellular domains, enabling the Notch-induced transcriptional programme. To evaluate the effect of full loss of endothelial Notch signalling, we induced Rbpj deletion in the ECs of mice carrying the alleles Rbpj floxed[24], Cdh5-CreERT2[25], and iSuRe-Cre. After tamoxifen induction, full deletion of the Rbpj gene occurs in MbTomato+ cells (Supplementary Fig. 1c–e). Rbpj gene deletion in most retina ECs from P1 to P6 induced an increase in vascular surface density and sprouting; however, at the same time it significantly decreased the total number of ECs at the angiogenic front (Fig. 1a–d). These results indicate that an increase in vascular density and sprouting can be accompanied by a significant decrease in the number of ECs generated, ultimately reducing vascular progression and angiogenesis (Fig. 1e). Interestingly, VEGF injection in the retina vitreous was previously shown to induce vascular expansion, through a process that is independent of its effect on EC proliferation[26].

So far it was not possible to assess the cell autonomous and long-term consequence of Notch LOF or gain-of-function in embryonic ECs in vivo, because complete disruption or activation of Notch signalling in blood vessels strongly affects vascular development and the physiology of the surrounding tissue, compromising embryonic development[14,15]. With this in mind, we used inducible fluorescent genetic mosaic mouse lines[13] that allowed us to interfere with Notch activity at single-cell resolution and analyse its impact on long-term EC proliferation and competition in an otherwise normal (wild-type) environment. These mouse lines are based on the Brainbow technology[27] and viral 2A peptide equimolar bicistronic gene expression[28]. In cells with Cre expression or activation of CreERT2, a stochastic and mutually exclusive recombination event occurs among the different LoxP sites, generating a fluorescent mosaic of cells with normal, low (DN-Maml1 or DN-Rbpj+), or high (NICD-PEST+) Notch activity (Fig. 1f and Supplementary Fig. 2). Unlike classical conditional knockout genetics, induction of genetic mosaics with the Tie2-Cre allele[29] in ECs at embryonic day (E) 8.5 was not embryonically lethal. This allowed us to track the fate and assess the relative proliferation and competitiveness of ECs with distinct Notch signalling levels over long periods, from E8.5 to postnatal stages. As expected, ECs with high Notch signalling levels had impaired proliferation and were gradually lost during vascular development. Unexpectedly, ECs with a cell-autonomous decrease in Notch signalling, which are considered to have higher sprouting and proliferative activity[1,2,30], did not win the vascular growth competition and most vessels in mosaic animals at postnatal day 20 (P20) were formed from ECs with normal Notch levels (Fig. 1f and Supplementary Fig. 2).

### Notch signalling prevents the arrest of proliferating ECs.
To characterise the role of Dll4/Notch in EC proliferation at higher temporal resolution, we used a Dll4-blocking antibody[5]. In contrast to inducible genetic deletions (Dll4 or Rbpj genes) or general pharmacological Notch inhibitors (Dll4-Fc, DAPT or DBZ) used before[5,11,19,20,23], a high dose (10 mg/kg) of anti-Dll4 acutely, potently and sustainably inhibited the endothelial Dll4-Notch signalling in the retinal angiogenesis model (Supplementary Fig. 1f). In this way, we were able to analyse the expression of several cell-cycle markers in the nuclei (Erg+) of proliferating ECs from 12 to 72 h after signalling inhibition in vivo (Fig. 2 and

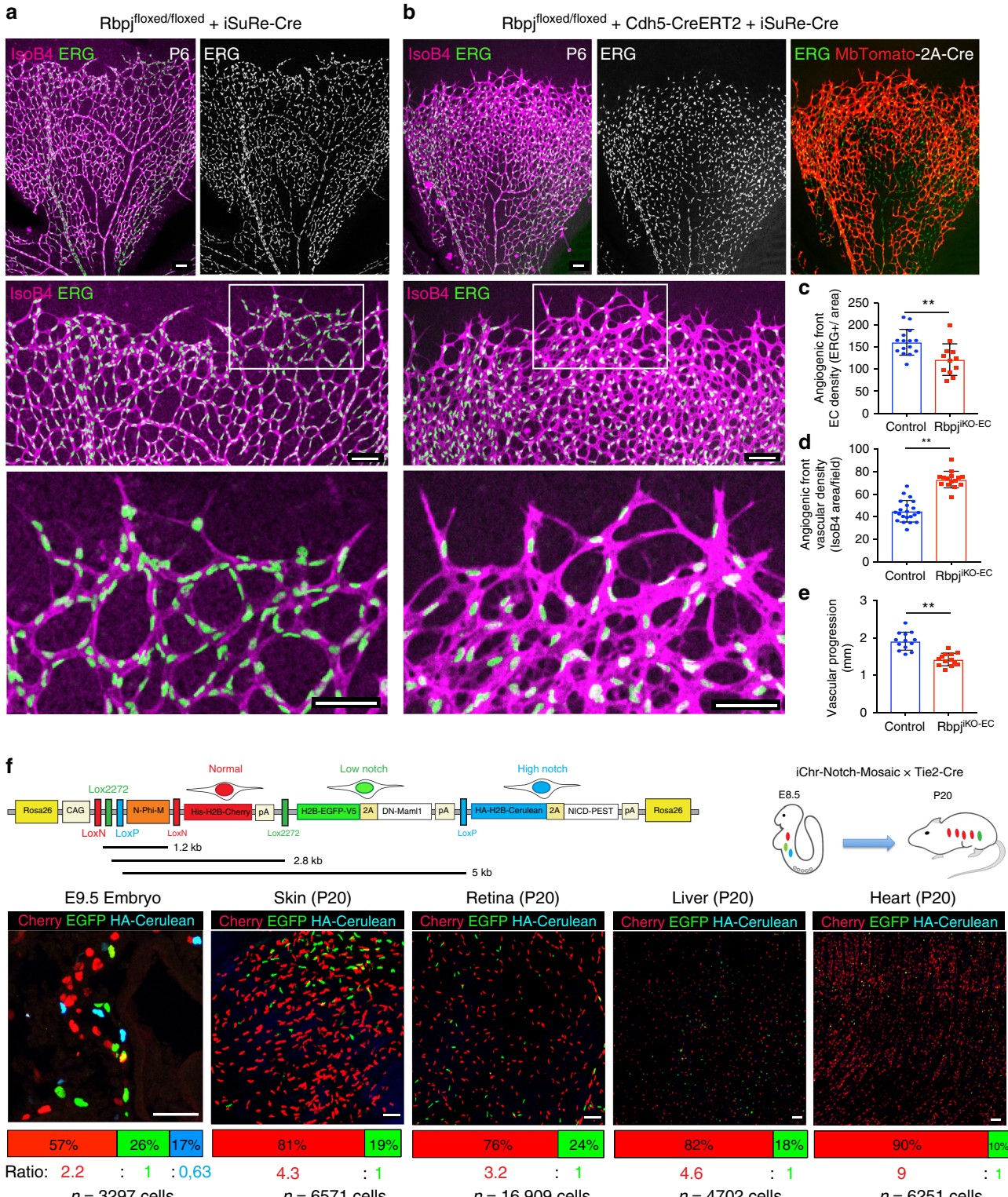

**Fig. 1** ECs with low- or high-Notch signalling are outcompeted during vascular development. **a**, **b** Confocal micrographs of the postnatal mouse retina vasculature showing that the full deletion of the *Rbpj* gene from P1 to P6 during retina angiogenesis, results in an increase in endothelial surface and sprouting (isolectinB4) and a decrease in the number of ECs (ERG+) and vascular progression. Cells with deletion of *Rbpj* from P1 to P3 are usually not found in arterial and peri-arterial endothelium at P6. See details of the *iSuRe-Cre* allele in Supplementary Fig. 1c–e. Scale bars, 80 μm. **c–e** Comparison of indicated parameters in large microscopic fields of control (*n* = 5) and mutant (*n* = 4) mice. **f** The *iChr-Notch-Mosaic* and *Tie2-Cre* mouse lines were crossed to generate fluorescent and genetic mosaics starting at E8.5 in growing ECs. Tissues of mice (*n* = 4) aged 20 days (P20) were analysed for the presence of Cherry, EGFP or HA-Cerulean cells. Comparison of the P20 ratios to the baseline/initial recombination ratio determined in E9.5 embryos (*n* = 4) or ES cells (see also Supplementary Fig. 2), shows that ECs with low Notch (EGFP+) or high Notch (HA-Cerulean+) are outcompeted during vascular development. Error bars indicate StDev; **\*\**p < 0.005. Two-tailed unpaired *t* test. Source data are provided as a Source Data file. Scale bars, 50 μm

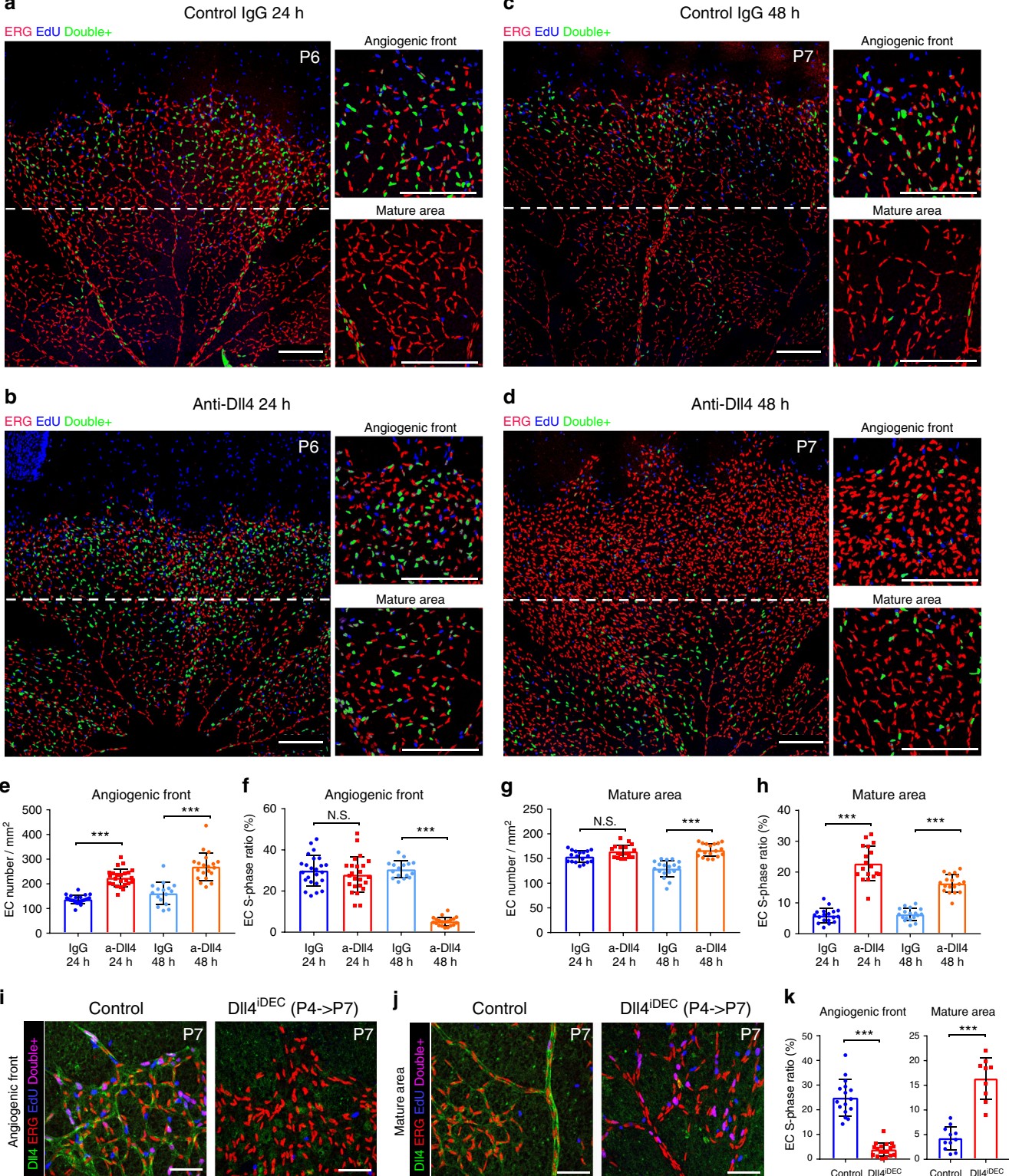

**Fig. 2** Dll4/Notch signalling inhibition induces context-dependent proliferative effects. **a–d** Confocal micrographs of the postnatal retinal vasculature from animals treated at P5 with IgG (control) or anti-Dll4 for 24 h (**a**, **b**) or 48 h (**c**, **d**). Anti-Erg (red) labels EC nuclei. EdU labels the nuclei of all cells in S-phase in the previous 4 h. Blue nuclei mark non-endothelial cells in S-phase, and double-positive (Erg+/EdU+) cell nuclei are pseudocoloured green to better highlight ECs in S-phase. Scale bars, 200 μm. **e–h** Comparison of indicated parameters in large microscopic fields of antibody treated retinas (*n* = 4 mice per group). Inhibition of Dll4/Notch signalling with anti-Dll4 leads to fast (24 and 48 h) endothelial expansion (number of Erg+ cells per field) followed by premature cell-cycle exit at the angiogenic front (**e**, **f**) and cell-cycle re-entry by mature ECs (**g–h**), which are indicated by the changes observed in the frequency of Erg+ cells in S-phase (EdU+). **i**, **j** EC-specific deletion of the *Dll4* gene from P4 to P7, results in a similar proliferative response. Scale bars, 70 μm. **k** Comparison of indicated parameters in large microscopic fields of control (*n* = 3) and mutant (*n* = 3) mice retinas. Error bars indicate std. dev.; NS nonsignificant; ***p* < 0.0005. One-way ANOVA with Tukey's post hoc test (**e–h**) or two-tailed unpaired *t* test (**k**). Source data are provided as a Source Data file

Supplementary Figs. 3 and 4). This high-resolution comparative analysis showed that loss of Dll4-Notch activity for 24–48 h leads to a transient increase in EC number and density (Fig. 2a–e, g) followed by a pronounced cell-cycle arrest. The fast cellular expansion observed in angiogenic areas occurred without an increase in the proportion of EdU+/Erg+ or Ki67+/H2B-GFP+ proliferating ECs (Fig. 2a, b, f and Supplementary Fig. 4 (24 h), and Supplementary Fig. 3g, h (12 h)). At the angiogenic front (AF), 96% of ECs were cycling (Ki67+/H2B-GFP+), and this frequency did not increase 24 h after Notch inhibition; rather, it decreased slightly (Supplementary Fig. 4b, f). The fast increase in EC number and density, without an increase in the frequency of proliferating cells (EdU+/ERG+ or Ki67+/ERG+), can only be explained by an increase in cell-cycle speed or a decrease in EC apoptosis. Since we have not detected apoptosis in any EC at the AF, these results suggest that Notch inhibition at the AF increases the speed of the EC cycle and not the proportion of proliferating ECs, in contrast to previous reports[4,5,11,20,22,23,31]. Importantly, this burst in vascular expansion was transient, lasting less than 48 h, by which time most ECs with loss of Notch signalling have already exited the cell cycle (Fig. 2d, f and Supplementary Fig. 4d, f). In contrast, in vessels of control IgG treated animals, the slower-dividing ECs, with normal Notch levels, continued to cycle for longer (Fig. 2c, f and Supplementary Fig. 4d, f), generating more ECs overtime (more ERG+ ECs per IsolectinB4 area) when compared with retinas treated with the Dll4-blocking antibody for 72 h (Supplementary Fig. 3a, b), mimicking the effect of *Rbpj* deletion from P1 to P6 (Fig. 1a, b). These results show that physiological Dll4/Notch activity slows EC proliferation in order to prevent the premature expansion and exhaustion of cycling ECs at the AF. Interestingly, Notch inhibition had a distinct effect in non-proliferative and more mature vessels, which are further from the source of angiogenic growth factors like VEGF. In quiescent (G0, Ki67−) retinal vessels, Dll4/Notch inhibition induced cell-cycle entry (Fig. 2h and Supplementary Fig. 4c, e, g), which is consistent with the previously described role of Notch as a suppressor of EC proliferation in mature quiescent adult vessels[3]. Deletion of *Dll4* in ECs (*Dll4^iDEC*) from P4 to P7 resulted in the same differential effect on the proliferation of angiogenic and mature ECs (Fig. 2i–k). Thus, our results indicate that if ECs are angiogenic at the time of Dll4/Notch LOF, they will proliferate faster but for a very short period, after which they exit the cell cycle; however, if ECs are quiescent, they reenter the cell cycle and divide for an extended period of time. This mislocalization of EC proliferation after Notch inhibition, from the AF to the more mature vascular area, impairs the growth of vessels into VEGF rich and hypoxic tissues (Fig. 1e).

**High-ERK activation induces EC cycle arrest**. We next investigated the impact of Notch on the activity of the MAPK/ERK pathway in proliferating (Ki67+) ECs. ERK signalling is known to induce both EC sprouting and proliferation[32–34]. We detected high levels of phospho-ERK (P-ERK) in tip cells (Fig. 3a left), the cells with the highest VEGF signalling input. In the absence of Dll4/Notch signalling for 24 h, P-ERK levels increased significantly in stalk cells and also in more distal quiescent ECs that are normally exposed to much lower VEGF concentrations (Fig. 3a, b and Supplementary Fig. 5a). Notch thus appears to suppress endothelial MAPK/ERK activity in both proliferative and quiescent ECs. This Notch-inhibition-induced increase in P-ERK levels is consistent with the observed cell-cycle re-entry of mature ECs, but not with the cell-cycle exit of angiogenic ECs observed in the retina. We therefore hypothesised that high mitogenic signalling (P-ERK) at the AF, caused by Notch inhibition in a context of extant high VEGF signalling, could trigger

an autoregulatory cell-cycle exit. To test this hypothesis, we crossed *iMb-Vegfr2-Mosaic* mice[13] with *Cdh5(PAC)-CreERT2* mice, allowing us to conditionally activate or inhibit Vegfr2 and ERK signalling at single-cell resolution and in a mosaic fashion in the embryo or in angiogenic retinal ECs (Fig. 3c–f). In these mice, ECs expressing MbTomato express a constitutively active form of VEGFR2[35], which significantly increases P-ERK activity (Fig. 3d and Supplementary Fig. 5b); in contrast, cells expressing MbYFP express a dominant-negative tyrosine kinase mutant VEGFR2 (VEGFR2^TkMut), which impairs VEGFR2 signalling[13]. In *iMb-Vegfr2-Mosaic* animals, the recombination is biased towards the first *MbTomato-VEGFR2^Ac.* cassette, since the LoxP2 and LoxP3 sites are separated by a large genetic distance, resulting in the lower frequency of MbYFP+ and MbKate2+ cells (Fig. 3c). To assess the relative proliferation of these cells shortly after genetic induction, we pulsed Cdh5-CreERT2+ ECs with tamoxifen at P3 and waited 3 days. Similarly to ECs lacking Notch signalling, ECs with higher VEGFR2 activity and P-ERK levels (Fig. 3d) exited the cell cycle (Fig. 3e–g). Most single ECs with high (MbTomato+) or low (MbYFP+) VEGFR2 signalling did not divide, and a smaller fraction completed the ongoing cell cycle to form 2-cell clones (Fig. 3h, i). Consistent with these observations, increasing VEGFR2 signalling in the developing retinal vasculature resulted in a significant decrease in angiogenesis (Fig. 3j, k). Thus, abnormally high mitogenic signals (P-ERK), induced either by increased VEGF signalling or decreased Notch signalling, arrest angiogenesis.

Our results could also suggest that the increase in P-ERK levels, after Notch inhibition, might be dependent on a cell-autonomous increase in Vegfr2 expression and signalling, as previously suggested[1,32,36,37]. However, Vegfr2 expression is not altered in ECs after Notch LOF in vivo (Supplementary Fig. 6a, b), and cells expressing a dominant-negative tyrosine kinase mutant VEGFR2 (VEGFR2^TkMut), which impairs VEGFR2 signalling and EC proliferation, can reenter the cell cycle and expand in the absence of Dll4/Notch signalling (Supplementary Fig. 6c, d), in agreement with other previous findings[13,19]. These results suggest that VEGF and Notch activities can both impact P-ERK levels and EC proliferation, independently of each other. However, at the AF both pathways are active, and therefore the overall effect on ERK signalling and EC proliferation of one pathway depends on the activity of the other.

**Tip cell fate mapping shows their low-proliferative capacity**. Tip cell specification is thought to be induced by the combination of high-VEGF and low-Notch signalling in a subset of ECs at the AF[11,18]. This process is transient, not fixed, allowing the reiterative sprouting and branching of other cells[37]. This implies that, at any given moment, cells at the tip position may be outcompeted by adjacent cells that more strongly sense or respond to VEGF or other stimuli. Tip cell position may therefore not accurately reflect the cell's activation status. To label with high resolution only those ECs that received a high-VEGF signalling input at the AF, and to fate-map their progeny, we generated *Gt (Esm1)^tm1(HA-H2B-Cerulean-2A-iCreERT2)* mice by CRISPR/Cas9-assisted gene targeting in mouse ES cells (Supplementary Fig. 7a). Unlike most genes expressed by angiogenic vessels, *Esm1* is strongly and exclusively expressed in endothelial tip cells[38], probably reflecting a high threshold for transcriptional activation. As expected, at any given moment, only a subset of tip cells experienced high-VEGF signalling and showed detectable HA-H2B-Cerulean expression (Fig. 4a). Consistent with the results presented in the previous section, only 31% of the Esm1+ tip cells were cycling (Ki67+), contrasting sharply with adjacent angiogenic stalk cells, which were 96% Ki67+ (31% Fig. 4a vs. 96%

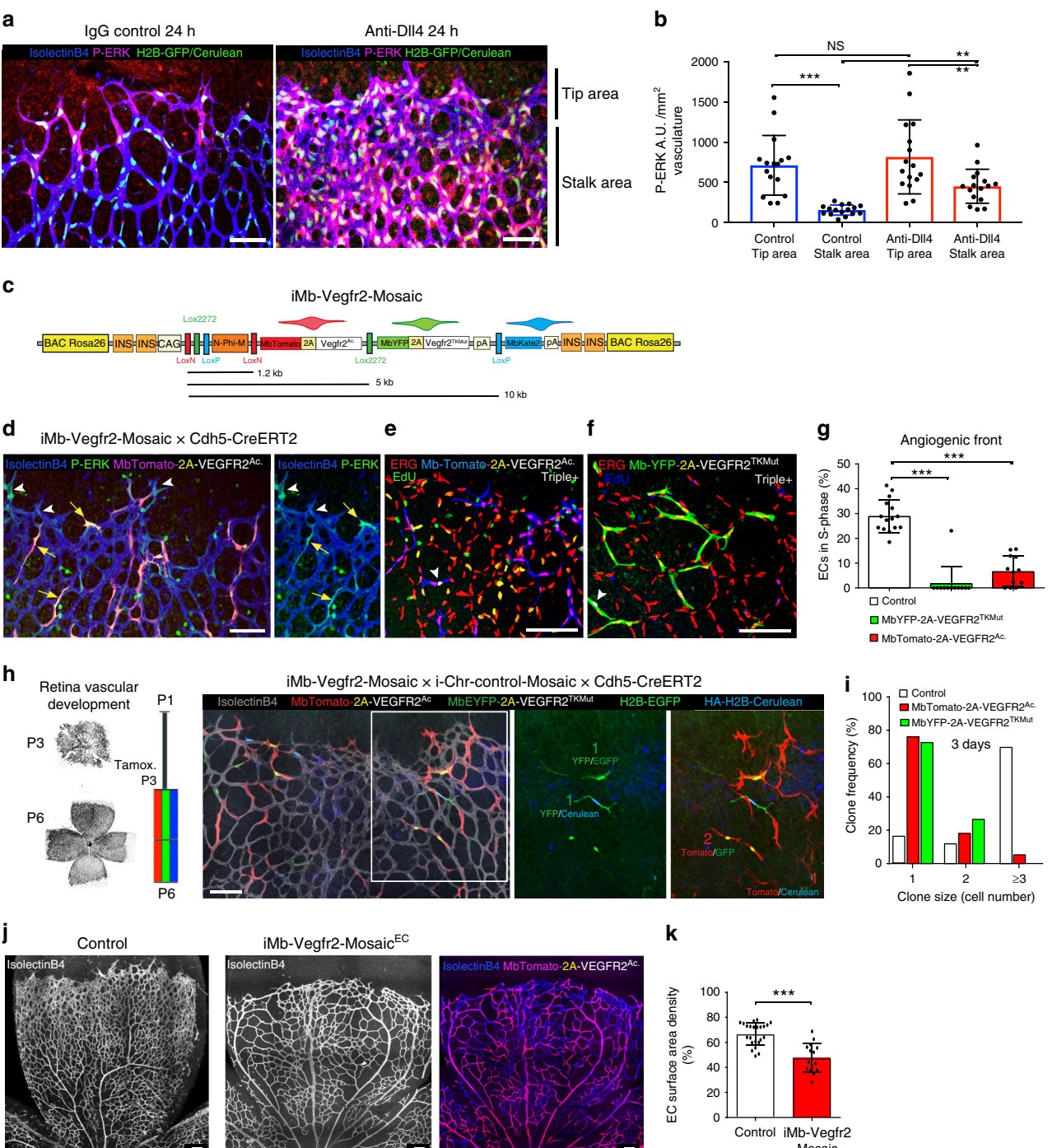

**Fig. 3** Angiogenic ECs with high VEGFR2 and ERK signalling exit the cell cycle. **a** Confocal micrographs of the retinal angiogenic front in *iChr-Cerulean/GFP x Tie2-Cre* reporter mice (see also Supplementary Fig. 4a), showing labelling of EC nuclei (Cerulean/GFP+). Activated ERK levels (P-ERK, red) are high in the nuclei and cytoplasm of endothelial tip cells (left image). Dll4/Notch inhibition increases P-ERK levels, particularly in stalk cells. **b** Quantification of P-ERK signals in tip and stalk cells from animals 24 h after injection with control IgG or anti-Dll4. Comparison of indicated parameters in large microscopic fields of IgG control ($n = 3$) or anti-Dll4 ($n = 4$) treated mice. **c** *iMb-Vegfr2 Mosaic* construct with genetic distances in kilobases (kb). **d** ECs expressing VEGFR2[Ac] (MbTomato+, yellow arrows) have high P-ERK levels, like some endothelial tip cells (white arrowheads). See also Supplementary Fig. 5b. **e**–**g** ECs expressing VEGFR2[Ac] (MbTomato+) or VEGFR2[TKMut] (MbYFP+) exit the cell cycle. Only a small fraction of MbTomato+ or MbYFP+ ECs are in S-phase (Erg+/EdU+-white nuclei), indicated with white arrowheads. Chart shows quantification of several microscopic fields from retinas of different mice ($n = 5$). **h, i** Most single ECs with very high (MbTomato+) or low (MbYFP+) VEGFR2 signalling do not divide over a 3 day period. The *iChr-Control-Mosaic* allele was used to increase single-cell pulse–chase clonal resolution. Chart shows quantification of several clones in large microscopic fields ($n = 4$ retinas per group). **j, k** The increase in VEGFR2 activity in *iMb-Vegfr2-Mosaic Cdh5-CreERT2* retinal vessels from P2 to P6 arrests physiological angiogenesis. MbTomato labelling shows the mosaic induction of the transgene in the endothelium (isolectinB4+), resulting in a significant decrease in vascular development. Chart shows quantification of large microscopic fields from several retinas of control ($n = 3$) and mutant ($n = 2$) mice. Scale bars in all panels, 100 μm. Error bars indicate std. dev.; NS nonsignificant; ***$p < 0.0005$; **$p < 0.005$. One-way ANOVA with Tukey's post hoc test (**b, g**) or two-tailed unpaired *t* test (**k**). Source data are provided as a Source Data file

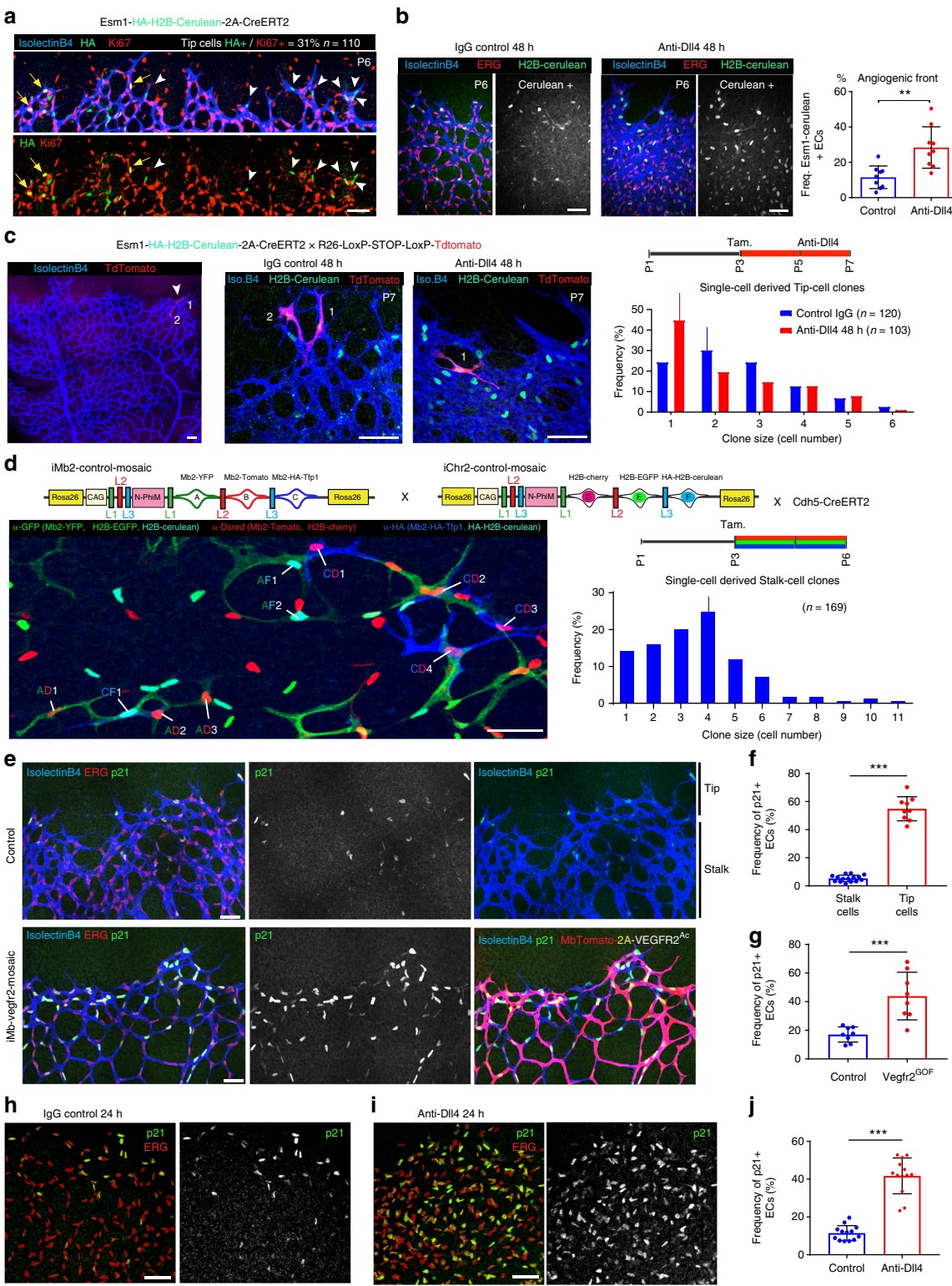

Supplementary Fig. 4f—IgG 24 h bar). Dll4/Notch signalling blockade induced a marked increase in the number of Esm1-expressing stalk cells (Fig. 4b), similar to our findings with P-ERK (Fig. 3a). Thus, while VEGF induces high P-ERK and Esm1 levels in tip cells, in stalk cells this induction is suppressed by the higher Notch activity in these cells.

To genetically pulse-label single endothelial tip cells and track their proliferative dynamics, we crossed *Esm1-HA-H2B-Cerulean-*

*2A-iCreERT2* and *Rosa26-LSL-TdTomato*[39] reporter mice (Fig. 4c) and induced recombination with tamoxifen at P3. This resulted in very few recombined clones per retina arising from a few Esm1+ tip cells (Fig. 4c left and Supplementary Fig. 7b), enabling us to score and assign single-tip-cell-derived clones over a 4-day period. To pulse single-endothelial stalk cells at very high-clonal resolution, we crossed *iMb-Control Mosaic* and *iChr-Control Mosaic* mice with *Cdh5(PAC)-CreERT2* mice (Fig. 4d and

**Fig. 4** Tip cells with endogenous high VEGF activity proliferate less than stalk cells. **a** Confocal micrographs showing that the *Esm1-HA-H2B-Cerulean-2A-iCreERT2* allele is expressed by a subset of endothelial tip cells in growing retinal vessels at P6. Only 31% of these cells are cycling (KI67+), contrasting with the adjacent endothelial stalk cells (96% cycling, see Supplementary Fig. 4b and f). **b** Inhibition of Dll4/Notch signalling upregulates the expression of *Esm1-HA-H2B-Cerulean* in ECs (Erg+) at the angiogenic front. Chart shows quantification of several microscopic fields from control ($n = 3$) and mutant ($n = 3$) mice. **c** In mice containing the *Esm1-HA-H2B-Cerulean-2A-CreERT2* and *Rosa26-LSL-TdTomato* alleles, the very low-recombination frequency allows individual endothelial tip cells to be genetically pulse–chased and their progeny followed over a few days. Blockade of Dll4/Notch signalling with Anti-Dll4 decreased the proliferation or clonal expansion of Esm1+ tip cells. The line above the bar indicates the most frequent clone size. Number of clones quantified is indicated. Please see also Supplementary Fig. 7b. **d** High-resolution *Dual ifgMosaic* labelling (*iMb2-Mosaic × iChr2-Mosaic*) and clonal analysis of single Cdh5-CreERT2+ stalk cells (see also ref. [13]). Recombination was induced with tamoxifen at P3 and vessels were analysed at P6. The histogram shows the frequency of identified dual-labelled cell clones according to their size. Total number of clones quantified is indicated. The line above the bar indicates the most frequent stalk-cell dual-labelled clone size (four cells). Please see also Supplementary Fig. 7c. **e–g** Confocal micrographs showing p21 protein in sprouting tip cells and very few stalk cells during normal retinal angiogenesis. ECs with high-VEGFR2 signalling (MbTomato+) strongly upregulate p21. Charts show quantification of several large microscopic fields from 4 (f) or 2 (g) full retinas per group. **h–j** Dll4/Notch signalling blockade for 24 h induces a strong increase in the frequency of p21-expressing ECs ($n = 3$ mice per group). Scale bars in all panels, 50 μm. Error bars indicate std. dev.; ***$p < 0.0005$; **$p < 0.005$. Two-tailed unpaired $t$ test. Source data are provided as a Source Data file

Supplementary Fig. 7c). Individual Esm1+ tip cells produced on average fewer progeny than individual stalk cells. In most cases, the Esm1+ tip cell pulse generated 2-cell clones after 4 days (Fig. 4c, blue bars), whereas a Cdh5+ stalk-cell pulse tended to generate 4-cell clones after only 3 days (Fig. 4d). This shows that even though tip cells are exposed to more VEGF, on average they proliferate significantly less, presumably because they exit the cell cycle or are arrested; in contrast, the stalk cells proliferate and form larger clones. Interestingly, when Dll4/Notch signalling was blocked, Esm1+ tip cells proliferated even less and most clones were formed by only 1 cell, instead of 2 cells (Fig. 4c). This result is consistent with a model where during the 4 days of the pulse–chase genetic experiment, non-proliferative tip cells can become proliferative stalk cells, but Dll4/Notch blockade inhibits this phenotypic switch, preventing these initially Esm1+/KI67− tip cells from reentering the cell cycle and adopting a clonal expansion mode typical of stalk cells.

Taken together, our results suggest that the Notch signalling effect on the endothelium is highly context dependent and is an essential element in maintaining a mitogenic balance that oscillates between sustained vascular proliferation in stalk cells and cell-cycle exit in sprouting tip cells. Notch blockade in VEGF-activated, high P-ERK, Esm1+ tip ECs further blocks their expansion, presumably by preventing tip-to-stalk cell transition. In contrast, blocking Dll4/Notch in proliferating (Ki67+, Esm1−) retinal stalk cells, which have relatively low-P-ERK levels, leads to an initial acceleration of the cell cycle and proliferation, but after 48 h these cells also exit the cell cycle (Fig. 2d and Supplementary Fig. 4d), a characteristic of sprouting tip cells. In mature quiescent retinal ECs (Ki67−), Dll4/Notch blockade induces a milder increase in P-ERK levels (compare Supplementary Fig. 5a chart with Fig. 3b chart), which is productive and sufficient to induce cell-cycle entry (mature area in Fig. 2b, d and h).

**p21 is abundant in tip cells and regulated by VEGF/Notch.** The data shown above suggest that during normal vascular development high-ERK activity is achieved only in the presence of high-VEGF and low-Notch signalling, a characteristic of tip cells (Fig. 3a left). To determine the molecular profile of angiogenic ECs with different Notch and VEGF signalling levels, we profiled gene expression in FACS-sorted AF retinal ECs from animals injected with control IgG, anti-Dll4 or anti-VEGF. Inhibition of Dll4/Notch signalling significantly increased expression at the AF of genes enriched in tip cells (*Esm1*, *Angpt2*, *Apln*) and the cell-cycle inhibitor *Cdkn1a* (p21), whereas inhibition of VEGF signalling had the opposite effect (Supplementary Fig. 6a, b). Our data above show that VEGF high/Esm1+ endothelial tip cells exit the cell cycle more frequently than stalk cells (69% of tip cells are

KI67− in Fig. 4a vs. 4% stalk cells KI67− in Supplementary Fig. 4f). We also found that a very high p21 protein content was more frequent in tip cells, particularly Esm1+ tip cells (Fig. 4e top panel and f), which are the cells with the highest P-ERK levels (Fig. 3a left). To confirm the direct link between VEGF/ERK signalling and p21, we analysed p21 expression in *iMb-Vegfr2-Mosaic* vessels, in which P-ERK is elevated in all MbTomato-2A-Vegfr2$^{Ac.+}$ cells (Fig. 3d and Supplementary Fig. 5b). Cells with very high-VEGFR2/ERK signalling strongly upregulated p21 (Fig. 4e, g) and exited the cell cycle (Fig. 3e–g). Similarly, after Dll4/Notch inhibition for just 24 h, p21 protein levels increased significantly in stalk ECs (Fig. 4h–j), again suggesting that these stalk ECs adopt a tip-like phenotype in the absence of Notch signals and activate the expression not only of tip cell markers (*Esm1*, *Angpt2*, *Apln*), but also of a cell-cycle inhibitor that is normally highly expressed only by sprouting tip cells. Previous studies with EC lines suggested that p21 might positively regulate EC proliferation[40,41]. We detected elevated p21 in KI67+ and P-ERK− high stalk ECs 24 h after Dll4/Notch inhibition (compare Fig. 4i with Supplementary Fig. 4b, f); however, at 48 h (Fig. 2d and Supplementary Fig. 4d, f), most of the cells that were previously cycling (Ki67+) and p21+, had exited the cell cycle. This finding indicates that the p21 upregulation observed after the inhibition of Dll4/Notch signalling precedes cell-cycle exit, consistent with its role as a cell-cycle inhibitor.

**Single ECs with Notch LOF often upregulate p21 and sprout.** Similar to the effect of Dll4/Notch signalling inhibition for 24 h, sustained *Rbpj* deletion in angiogenic retinal ECs from P1 to P6 also resulted in strong increases in P-ERK and p21, coinciding with a marked arrest of EC proliferation and angiogenesis, even though this was accompanied by increased vascular density and hypersprouting (Fig. 5a–d).

By inducing single-cell mutant mosaics in *Cdh5-CreERT2 × iSuRe-Cre × Rbpj^{floxed}* or *iChr-Notch Mosaic* mice, we determined that the p21 upregulation and cell-cycle exit after Rbpj/Notch signalling loss is cell-autonomous, increasing in frequency with the proximity to the VEGF source. *Rbpj* deletion, or expression of DN-Maml1, upregulated p21 in individual ECs at the AF, particularly at the leading edge, but not in the more mature vascular area (Fig. 5e–h and Supplementary Fig. 8a–d). In addition, single ECs with Rbpj/Notch signalling loss and higher p21 expression were found more frequently at the leading edge of the vascular network (Fig. 5h and Supplementary Fig. 8e), suggesting that p21+ and arrested (KI67−) ECs have higher migratory activity and occupy the tip position more frequently than the adjacent proliferative (KI67+/p21−) cells.

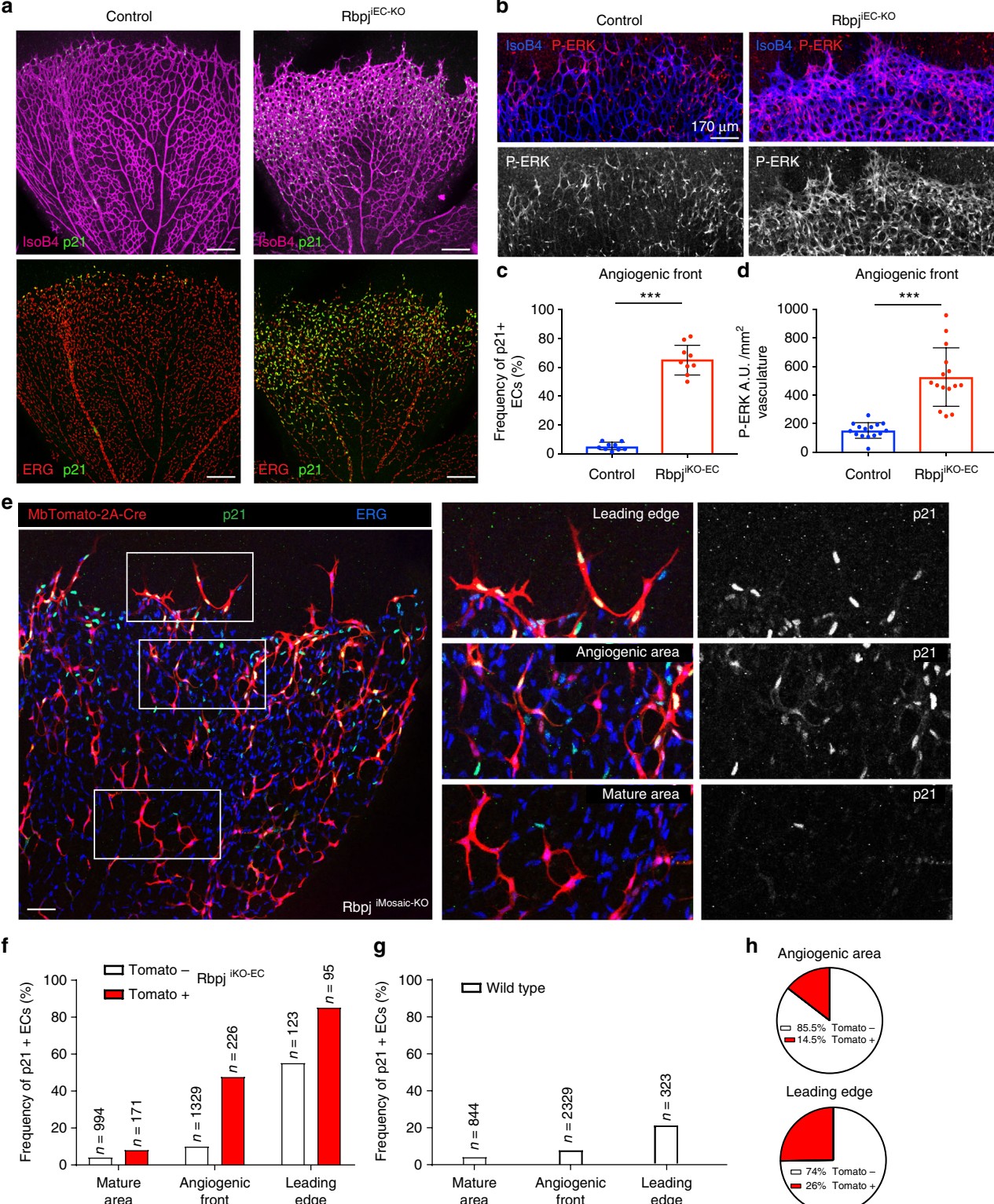

**Fig. 5** Individual ECs with loss of Rbpj/Notch signalling often express p21 and occupy the leading edge. **a**–**d** The induction of *Rbpj* genetic deletion from P1 to P3 results in a strong increase of p21 and P-ERK levels at the AF in P6 retinas. Charts show quantification of several large microscopic fields (*n* = 3 mice per group). Scale bars, 200 μm. **e** Confocal micrographs of P6 retinas from animals with inducible mosaic deletion of *Rbpj* in ECs (*Rbpj*^iECMosaic-KO^) expressing MbTomato-2A-Cre. Images to the right are high-magnification views of boxed areas, showing the indicated signals in more detail. Scale bar, 115 μm. **f**, **g** Quantification of mutant (Tomato+) and control (Tomato−) ECs (ERG+) from *Rbpj*^iECMosaic-KO^ retinas (*n* = 6). Individual cells without Rbpj/Notch signalling (MbTomato-2A-Cre+) are more frequently p21+, especially if they are at the leading edge of retina vessels. In the mature vascular area, most ECs with *Rbpj* deletion do not upregulate p21. Note that a fraction of Tomato− cells may have *Rbpj* deletion (compare white bars in **f** with **g**). **h** Cells without Rbpj/Notch signalling (MbTomato-2A-Cre+), are more frequently found at the leading edge of vessels than at the angiogenic front. Error bars indicate std. dev; ***$p$ < 0.0005. Two-tailed unpaired $t$ test. Source data are provided as a Source Data file

These results suggest the existence of a cell-intrinsic auto-regulatory mechanism, regulated by physiological Notch levels, that maintains endothelial stalk cell identity and proliferation by repressing overactivation of ERK, and thus the expression of the cell cycle inhibitor p21, both of which are upregulated in actively sprouting tip cells. Interestingly, only a fraction of single-angiogenic ECs with full Dll4/Notch or *Rbpj* LOF had high p21 levels at any given timepoint of analysis (Fig. 4i, j Anti-Dll4 (24 h) and Fig. 5f *Rbpj* deletion (P1–P6)), suggesting single-cell heterogeneity in p21 upregulation. This heterogeneity, may be related with the known heterogenous and highly dynamic response of ECs to VEGF signalling[37], and is also evident in the Esm1 reporter and P-ERK immunostaining analysis presented above (Fig. 4a, b and Fig. 5b).

**ERK regulates p21 to balance EC sprouting and proliferation.** Our results show that EC p21 expression in vivo coincides with increased P-ERK, which can be induced either by high VEGF signalling or by Notch signalling inhibition in the context of active angiogenesis; moreover, soon after upregulating P-ERK and p21, most of these ECs exit the cell-cycle. Weak ERK activation is associated with an increase in mitogen-induced cell proliferation[42]; however, strong ERK activation was previously linked with cell-cycle arrest and p21 upregulation in established cell lines in vitro[43,44]. In agreement with this, P-ERK levels are low in proliferating endothelial stalk cells, and high in sprouting tip cells. To determine the functional link between ERK activity, p21, and EC proliferation we examined the effect of the MEK/ERK inhibitor (SL327) during retinal angiogenesis. Similarly to previous results obtained in zebrafish embryos[33], ERK activity inhibition for 24 h prevented EC sprouting; moreover, it also significantly diminished the frequency of angiogenic and mature ECs in S-phase, particularly after anti-Dll4 treatment (Fig. 6a–e). ERK activity inhibition for 24 h also significantly reduced p21 upregulation in anti-Dll4 treated retinas, although p21 protein remained in some endothelial tip cells (Fig. 6f, g), suggesting slow p21 protein turnover. These results indicate that while strong ERK activation as occurs in tip-ECs induces p21 expression and sprouting, the weaker ERK activity in stalk-ECs is unable to induce p21 expression, and instead sustains EC proliferation.

**p21 specifically arrests ECs with high mitogenic stimulation.** Given the strong regulation of p21 by VEGF, Notch and ERK stimuli, we next investigated the role of p21 in angiogenesis. Interestingly, animals with full p21 LOF displayed no significant defects in retinal angiogenesis or in the frequency of proliferation by stalk-ECs with physiological Notch signalling (Fig. 7a, c, d), likely because p21 expression is undetectable in most of these cells (Fig. 7e, f). In contrast to stalk cells, 70% of Esm1-Cerulean+ endothelial tip cells were p21+ (Fig. 7e, f) and KI67− (Fig. 4a).

In the absence of p21 function, the frequency of arrested Esm1-Cerulean+ endothelial tip cells decreased to 40%, whereas the frequency of cycling stalk-cells remained the same (Fig. 7g, h). These results indicate that p21 inhibits the proliferation of tip cells but not stalk cells, presumably because it is mainly expressed by VEGF-high, or Notch-low, Esm1+ tip cells. When Dll4/Notch signalling was inhibited in retinal vessels, p21 expression increased significantly in stalk cells (Fig. 4i). In this context of p21 upregulation, we observed that p21 loss partially inhibited the cell-cycle arrest observed after Dll4/Notch blockade (Fig. 7b, c), leading to a greater increase in EC number and density (Fig. 7d). These results indicate that the p21 upregulation observed in tip cells, or after Dll4/Notch blockade (Fig. 4h–j), is one of the drivers of cell-cycle arrest of these cells, even though other cell-cycle

regulators and checkpoints may partially compensate the full loss of p21 function.

## Discussion

Pharmacological compounds that inhibit or promote the function of the VEGF and Notch pathways have been widely used in the clinics. The currently prevailing view is that high VEGF signalling or Dll4/Notch blockade increases angiogenesis and EC proliferation in any physiological or pathological context[1,8,45,46]. Our study shows that the effects of VEGF, Notch, and ERK signalling are instead highly context-dependent. In the case of angiogenic and already activated ECs, the increase in EC proliferation observed after increasing VEGF or decreasing Dll4/Notch signalling is transient, and is followed by cell-cycle exit. Our study may also explain the failure of VEGF-delivery therapies to promote functional angiogenesis, for example those used to treat cardiovascular ischaemic disease[47]. According to our results, delivering additional VEGF will induce p21 and cell-cycle arrest in vessels close to the VEGF source while presumably increasing the activation of more distal vascular cells, which are the least important for effective vascularisation and irrigation of the ischaemic tissue.

In this study, we used a series of in vivo pulse–chase genetic and pharmacological experiments that provide a detailed analysis of the functional interactions between VEGF, Notch, ERK, and p21 in angiogenic and quiescent vessels. We show how these molecules act in different vascular contexts and how their levels need to be properly balanced in order to regulate EC proliferation and prevent premature cell-cycle arrest. Our results suggest that ECs have a bell-shaped dose-response to mitogenic stimuli (Fig. 8a), according to which a high-VEGF input, or low-Notch input, induces ERK phosphorylation and downstream signalling, which when high induces p21 and cell-cycle arrest, a process associated with effective endothelial sprouting and migration. Most sprouting retinal tip cells are Ki67−/Esm1+/p21+ and have high P-ERK levels (Fig. 8b). The induction of these molecular mechanisms by VEGF, is strongly counteracted by higher Notch activity in stalk cells. Therefore, the main role of physiological Notch levels at the AF is to supress ERK signalling, in order to maintain stalk-cell proliferation and prevent their premature arrest (Fig. 8c). The cell-cycle arrest induced by Notch inhibiton in proliferating stalk cells is dependent on the increase in ERK activity and p21 levels. In contrast to angiogenic vessels, mature and quiescent ECs have significantly lower VEGF and ERK signalling levels, and therefore the increase in ERK activity after Notch inhibition is more moderate and therefore productive, inducing cell-cycle entry (Fig. 8c).

Recent work suggested that some tip cells or tip-derived ECs may have high Notch and Cxcr4 signalling, which was found to be required for their migration toward arteries[48,49]. Our results suggest that most retina tip cells, particularly those with the highest VEGF signalling (Esm1-Cerulean+/p21+), have low-Notch signalling and migrate in the opposite direction. We also show that conversion of tip cells to proliferative stalk cells can be prevented by blocking Dll4/Notch activation, maintaining their excessive ERK and p21 expression and thus preventing the adoption of a more proliferative stalk cell phenotype. It is nonetheless possible that a fraction of tip cells has high Notch activity, or acquires it during the dynamic phenotypic inter-conversion of tip cells to stalk cells. This may induce reversal of EC polarity and the migration of tip cells away from the angiogenic front and towards the arteries, as proposed recently[48–50].

Contrasting our in vivo data, Notch signalling inhibition, or increased VEGF signalling in EC lines always increases their proliferation (Supplementary Fig. 1)[21,22], suggesting that the bell-

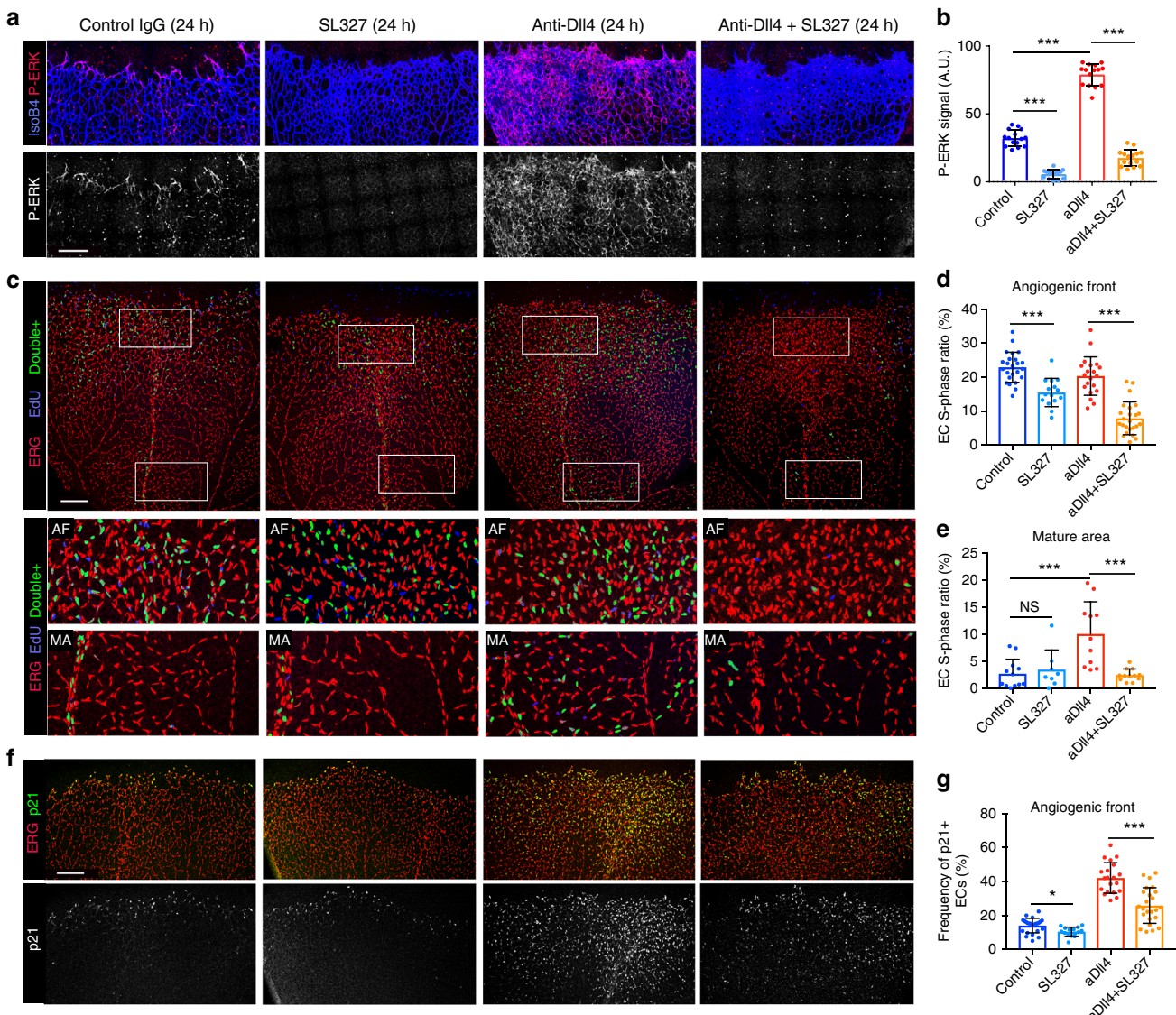

**Fig. 6** Inhibition of ERK phosphorylation decreases EC sprouting and proliferation. **a**, **b** Confocal micrographs of retinas treated with IgG (control) or anti-Dll4 antibody in the presence or absence of MEK/ERK inhibitor (SL327) for 24 h. P-ERK immunostaining is strongly reduced in SL327-treated retinas, leading to decreased sprouting (n = 3 mice/6 retinas per group). **c** Confocal micrographs of retinas treated with IgG (control) or anti-Dll4 antibody in the presence or absence of SL327 for 24 h and immunostained for ERG (red) and EdU (blue) to check effect of ERK signalling on EC proliferation. Dual ERG +/EdU+ cells are pseudocolored in green. **d**, **e** Inhibition of ERK phosphorylation with SL327 for 24 h, significantly reduces the frequency of EdU+/ERG+ ECs at the AF and the anti-Dll4 induced proliferation in the mature vascular area (n = 3 mice/6 retinas per group). **f**, **g** Confocal micrographs of retinas treated with IgG (control) or anti-Dll4 antibody in the presence or absence of SL327 for 24 h and immunostained for ERG (red) and p21 (green). Quantifications show that SL327 treatment for 24 h reduced the frequency of p21+ ECs, with a particularly strong inibitory effect on the p21 upregulation induced by anti-Dll4 (n = 3 mice/6 retinas per group). Scale bars in all panels, 200 μm. Error bars represent StDev. *p < 0,05, ***p < 0.0005 and NS not significant. One-way ANOVA with Tukey's post hoc test. Source data are provided as a Source Data file

shaped dose–response and cell-cycle arrest mechanism we identified here is inactive in vitro, or is bypassed. This discrepancy may reflect significant differences between the in vitro and in vivo biochemical and physical endothelial microenvironment, such as shear stress, oxygen concentration, EC-to-EC junctional stability and nutrient availability. All these factors might interact with and modify the operation of the cell-cycle arrest mechanism identified.

The role of Cdkn1a (p21) in angiogenesis and its regulation by Notch has mostly been studied in vitro. Notch activation in EC lines suppresses p21 expression, and since Notch overactivation is accompanied by cell-cycle arrest, p21 was proposed to be a positive regulator of EC proliferation[40]. This contrasts with the proposed interaction between Notch and p21 in keratinocytes, where Notch activation induces p21 expression to induce cell-cycle arrest[51]. Our results show that p21 plays a minor role during physiological angiogenesis, likely because under physiological Notch signalling, it is expressed only by endothelial tip cells, which constitute a very small fraction of the proliferating EC pool. However, when Notch signalling is inhibited, p21 expression and its importance for angiogenesis increase.

The expression of p21 was shown before to be triggered by many different pathways, including DNA-damage and p53, labelling cells undergoing senescence[52]. However, p53 expression did not change after Dll4/Notch inhibition (Supplementary Fig. 6). Even though we cannot entirely rule out a potential

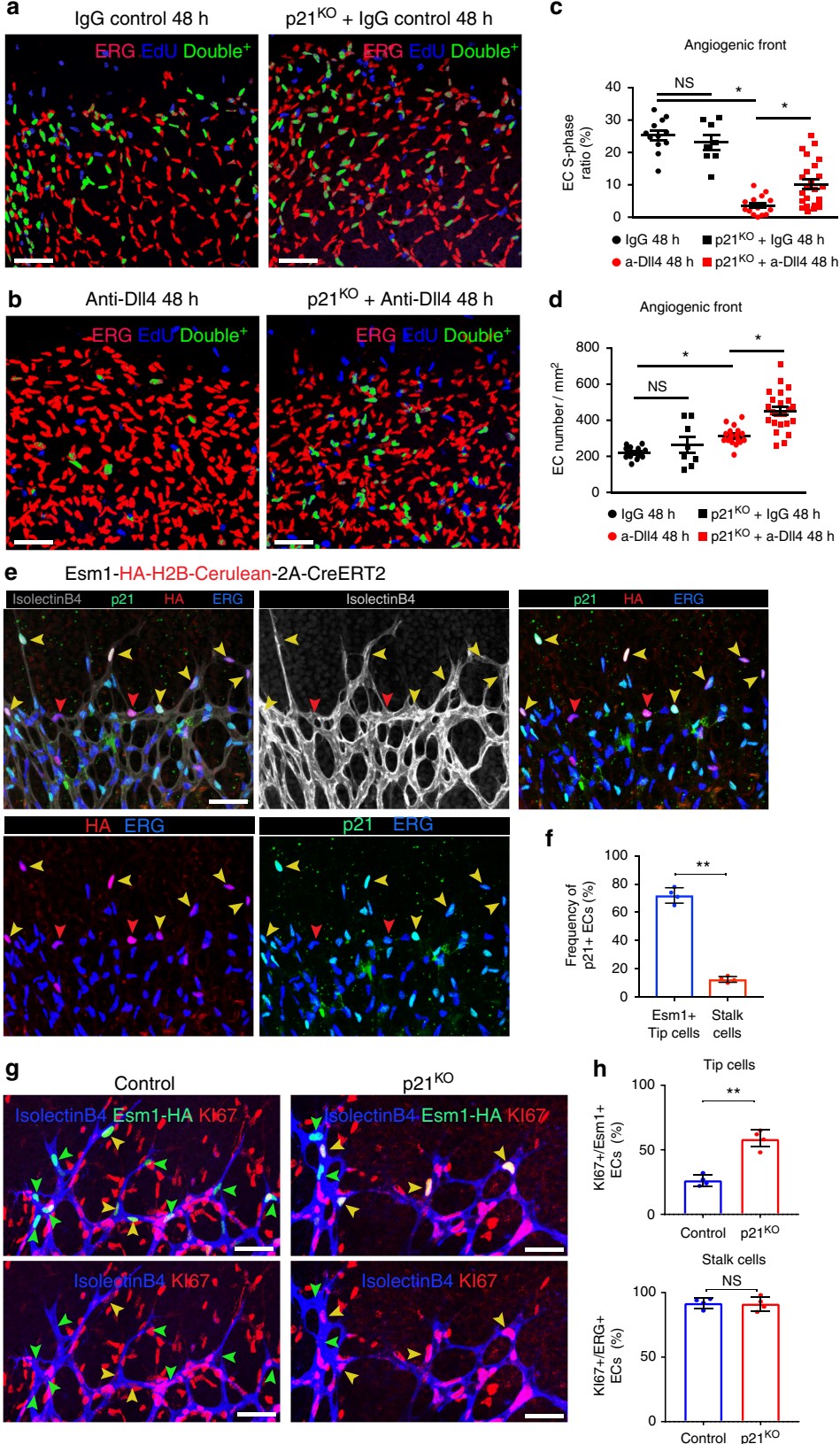

contribution of p53 and DNA damage for the observed upregulation of p21, our work shows that p21 expression can be instead directly activated by ERK signalling in ECs (Fig. 6), similarly to what was shown before in other cell types[53,54].

Here, we have used mainly the embryonic and postnatal retinal angiogenesis systems as experimental models because they permit a high level of genetic, cellular and temporal resolution. Given the context and dose-dependent signalling roles identified in this

**Fig. 7** p21 inhibits proliferation of ECs exposed to high-mitogenic stimulation. **a–d** ECs (ERG+ nuclei) in control and p21 knockout retinas (**a**) have similar frequencies of cells in S-phase (Erg+/EdU+ double-positive cells, pseudocoloured green). After administration of anti-Dll4 antibody for 48 h (**b**), the frequency of S-phase ECs decreased significantly in control retinas due to cell-cycle arrest. In p21 knockout retinas, anti-Dll4-induced cell-cycle arrest was less frequent (**c**), and resulting in the generation of more ECs (**d**). Charts show quantification of large microscopic fields from several control and mutant retinas (IgG 48 h, $n = 4$; p21$^{KO}$ IgG 48 h, $n = 4$; a-Dll4 48 h, $n = 5$; p21$^{KO}$ a-Dll4 48 h, $n = 6$). **e, f** Confocal micrographs of retinas from P6 Esm1-HA-H2B-Cerulean-2A-iCreERT2 mice ($n = 4$) and corresponding chart (**f**), showing that 70% of Esm1+ tip-ECs (HA+, red) are p21+ (green), as indicated by the yellow arrowheads. **g, h** Confocal micrographs of P6 retinas from control ($n = 4$) and p21$^{KO}$ ($n = 4$) mice carrying the Esm1-HA-H2B-Cerulean-2A-iCreERT2 allele, showing that p21 blocks tip-EC proliferation (yellow arrowheads indicate Ki67+, Esm1+ tip-ECs) but does not affect stalk-EC proliferation. Scale bars in all panels, 50 µm. Error bars represent SEM (**c, d**) or std. dev. (**f, h**). NS nonsignificant; $*p < 0,05$; $**p < 0.005$. One-way ANOVA with Tukey's post hoc test (**c, d**), two-tailed unpaired $t$ test (**f, h**). Source data are provided as a Source Data file

study, it is possible that an increase in VEGF, or decrease in Notch signalling, will trigger different outcomes in different vascular networks or pathological settings. Indeed, and in contrast to the Esm1+ retinal tip cells, which need to migrate very actively on a layer of hypoxic astrocytes, zebrafish intersegmental tip cells seem to be able to sprout and proliferate at the same time[4], presumably due to relatively lower-VEGF signalling or higher Notch signalling. We propose that this heterogeneity in the behaviour of tip, stalk and quiescent ECs, is induced by a bell-shaped dose–response to mitogenic/ERK stimuli that is highly dependent on the vascular and signalling context and oscillates between active endothelial proliferation and cell-cycle arrest (Fig. 8).

This study shows how critical the cellular context and the levels of mitogenic stimuli are for appropriate angiogenesis, where more can result in less. The identified dual, and often mutually exclusive, role of mitogenic stimuli levels on endothelial sprouting and proliferation will need to be considered in future pro-angiogenic and anti-angiogenic therapies.

## Methods

**Mice, genetic experiments and pharmacological inhibition**. To generate Notch and VEGFR endothelial-cell mosaics, we intercrossed the endothelial-specific Cre lines Tie2-Cre[29] or the tamoxifen inducible Cdh5(PAC)-CreERT2 mouse line[25] with the following ifgMosaic mouse lines:[13] iChr-Notch-Mosaic (Gt(Rosa)26Sor$^{tm1(iChr-Notch Mosaic)}$), iChr-Notch-V2-Mosaic (Tg(BAC Rosa26)$^{(iChr-Notch Mosaic-v2)}$) or iMb-Vegfr2-Mosaic (Tg(BAC Rosa26)$^{(iMb-Vegfr2 Mosaic)}$). To specifically label the nuclei of ECs in some experiments, we used the iChr-Cerulean/Gfp Mosaic (Gt(Rosa)26Sor$^{tm1(iChr-Cerulean/GFP/Kate2 Mosaic)}$) mouse line (Supplementary Fig. 2i–o). To increase the mosaic single-cell clonal resolution after recombination, we intercrossed the iMb2-Control-Mosaic (Gt(Rosa)26Sor$^{tm1(iMb2-Control-Mosaic)}$) and iChr2-Control-Mosaic (Gt(Rosa)26Sor$^{tm1(iChr2-Control-Mosaic)}$) mouse lines[13].

To recombine and fate-map endothelial tip cells experiencing high VEGF signalling, we generated the Esm1$^{tm(HA-H2B-Cerulean-2A-iCreERT2)}$ mouse line. By using CRISPR/Cas9-assisted homologous-dependent recombination in mouse ES cells, we inserted the HA-H2B-Cerulean-2A-iCreERT2 cassette in-frame with the endogenous Esm1 gene ATG initiation codon (Supplementary Fig. 5a). These gene-targeted ES cells were validated by Southern blot and later used to produce mice. This mouse line was later intercrossed with the Gt(Rosa)26Sor$^{tm14(CAG-LSL-tdTomato)Hze}$[39] reporter mouse line (abbreviated as Rosa26-LSL-TdTomato) to fate-map individual endothelial tip cells.

For experiments involving inducible and endothelial-cell specific genetic Notch signalling loss-of-function we used Dll4$^{floxed}$ mice[55] or Rbpj$^{floxed}$ mice[24] crossed with the Cdh5(PAC)-CreERT2[25] or iSuRe-Cre mouse lines (detailed description of this line will be provided elsewhere). All primer sequences required to genotype mice are provided in Supplementary Table 1.

To activate CreERT2 in pups, we administered 120 µg/g body weight of tamoxifen (T5648) or 20 µg/g of 4-OH tamoxifen (H6278) diluted in a 1:1:2 solution of EtOH:Cremophor:phosphate buffered saline (PBS)[56]. To investigate the function of p21 in angiogenesis, we used Cdkn1a (p21) knockout mice[57]. Dll4/Notch signalling inhibition in ECs was achieved by using the humanised phage antibody YW152F, developed by Genentech[5]. Human IgG (Sigma, I4506) was used in littermates as a control treatment. A single subcutaneous injection of 40 µl of IgG or anti-Dll4 (0.5 mg/ml in PBS) was administered at the indicated stage. Retinas were collected 12, 24, 48 or 72 h after blocking antibody administration and processed for immunohistochemistry. In some experiments, the Y-secretase inhibitor IX (DAPT, Calbiochem) was injected intraperitoneally (IP) at 100 mg/kg per day. To inhibit Notch signalling and ERK phosphorylation at postnatal stages anti-Dll4 (10 mg/kg) and SL327 (MEK

inhibitor from Selleckchem, at 120 mg/kg) were injected IP into mouse pups at P5 and again 16 h later, before collecting the tissues at P6 (injections at −24 and −8 h time points). To detect proliferating cells actively synthesising DNA, EdU (Invitrogen—A10044) was injected IP 4 h before sacrifice; the signal was developed with the Click-it EdU Alexa Fluor 647 Imaging Kit.

Experiments involving animals were conducted in accordance with institutional guidelines and laws, following protocols approved by local animal ethics committees and authorities (Comunidad Autónoma de Madrid and Universidad Autónoma de Madrid—CAM-PROEX 177/14 and CAM-PROEX 167/17).

**Immunohistochemistry**. For mouse retina immunostaining, eyes were collected at the indicated time points and fixed in 4% PFA in PBS for 1 h at room temperature (RT). After two PBS washes, retinas were micro-dissected and stained as described previously[18]. Briefly, retinas were blocked and permeabilized with 0.3% Triton X-100, 3% foetal bovine serum (FBS) and 3% donkey serum in PBS. Samples were then washed twice in PBLEC buffer (1 mM CaCl$_2$, 1 mM MgCl$_2$, 1 mM MnCl$_2$ and 1% Triton X-100 in PBS). Biotinylated isolectinB4 (Vector Labs, B-1205, diluted 1:50) or primary antibodies (see below) were diluted in PBLEC buffer and tissues were incubated in this solution for 2 h at RT or overnight at 4 °C. After five washes in blocking solution diluted 1:2, samples were incubated for 1 h at RT with Alexa-conjugated secondary antibodies (Molecular Probes). After two washes in PBS, retinas were mounted with Fluoromount-G (SouthernBiotech). To detect EdU-labelled DNA, an additional step was performed before mounting using the Click-It EdU kit (Thermo Fisher, C10340).

Primary antibodies were used against the following proteins: Erg (Abcam, ab110639 1:500); GFP/YFP/Cerulean (Acris, R1091P, 1:400); p21(Santa Cruz, sc-397-G, 1:100 and rat anti-p21 from CNIO-Centro Nacional de Investigaciones Oncológicas); Ki67 (Thermo Fisher, RM-9106-S0, 1:400); Cherry/Tomato (Clonetech, 632496, 1:400); pERK (Cell signalling, 4370S, 1:400). To combine multiple antibodies in the same immunodetection, in some instances the following proteins were detected with conjugated primary antibodies: GFP/YFP/Cerulean (AF-488, Invitrogen A213111, 1:100); Erg (AF-647, Abcam ab196149, 1:100); HA (AF-647, Cell Signalling 3444S, 1:100 or AF-594, Thermo Fisher A-21288 1:200); Tomato (CF-594, Biotium 20422, 1:400); and V5 (FITC, Thermo Fisher R963–25 1:100 or AF-488, Biorad MCA1360A488, 1:100). To detect two primary antibodies from the same host, we combined detection with conjugated Fab fragment secondary antibodies and directly conjugated primary antibodies. The following secondary antibodies were used: donkey anti-rabbit (Thermo Fisher, AF-488, A-21206 or AF-680, A-10043); donkey anti-goat (Thermo Fisher, AF-488, A-11055 or AF-633, A21082 or AF-647, A-21447 or AF-680, A-21447); donkey anti-rabbit Fab fragment (Jackson Immunoresearch, Cy3, 711-167-003 or AF-647 711-607-003); and streptavidin 405 (Thermo Fisher, S-32351, 1:200).

**Flow cytometry of ECs**. To detect, separate, and profile the different endothelial cell populations in ifgMosaic or iSuRe-Cre mice, we interbred these mice with Tie2-Cre or Cdh5-CreERT2 animals to activate recombination and fluorescent protein expression in the endothelium. Embryos or organs were collected in PBS, minced, and digested with 2.5 mg/ml type I collagenase (Thermo Fisher, 17100017), 2.5 mg/ml dispase II (Thermo Fisher, 17105041), and 50 ng/ml DNAseI (Roche) at 37 °C for 20 min to create a homogeneous cell suspension. Cell suspensions were passed through a 70 µm filter to remove any undigested tissue. To remove erythroid cells, cell suspensions and blood samples were incubated for 10 min on ice in blood lysis buffer (0.15 M NH$_4$Cl, 0.01 M KHCO$_3$ and 0.01 M EDTA in ddH$_2$0). Before analysis, cell suspensions were incubated at 4 °C for 30 min with APC rat anti-mouse CD31 (BD Pharmigen) diluted 1:200. The Flow cytometry analyses were performed either with a FACS Aria Cell Sorter (BD Biosciences) or a Synergy4L Sorter. Viable (DAPI-) ECs (APC-CD31+) were sorted and analysed according to their endogenous fluorescence (GFP, Cherry, Cerulean or MbTomato).

To isolate ECs specifically from the neonatal mouse retina AF, we first separated the mature and angiogenic-front vascular areas under a dissecting microscope. Tissues from these two distinct areas were then cut and processed for dissociation

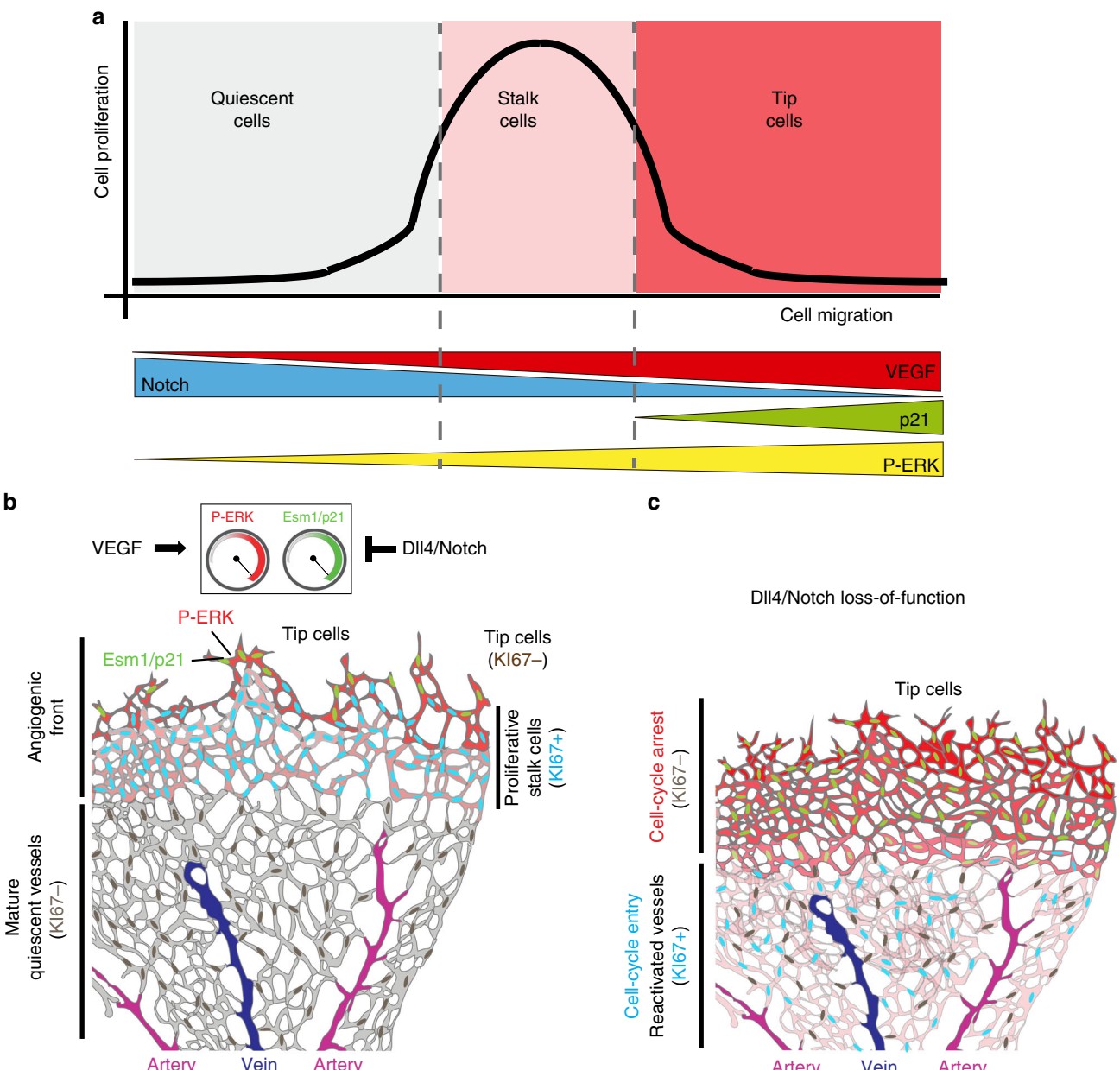

**Fig. 8** High-mitogenic stimulation arrests angiogenesis. **a** Proposed bell-shaped dose–response to mitogenic stimulation regulated by Notch, VEGF, ERK, and p21. At very low levels of VEGF signalling, ECs are quiescent, since they have Notch signalling, which supresses ERK activity and cell proliferation. Stalk cells have a properly balanced level of Notch and VEGF signalling, resulting in an ERK activity level ideal for cell proliferation. Tip cells have high-VEGF signalling, and low-Notch signalling, resulting in cumulative high ERK activity, which induces p21, cell-cycle arrest, and cell sprouting/migration. **b** Illustration showing the identified distribution of the indicated cellular markers in tip, stalk and quiescent ECs. Colours indicate P-ERK (red cytoplasm), Esm1/p21 (green nuclei), KI67 (blue if positive and dark grey if negative). Colour intensity indicates the observed strength of signal or expression. At the angiogenic front, endothelial tip cells receive a higher mitogenic stimulus from VEGF. This induces ERK activation and high expression of Esm1 and p21. Higher Notch signalling in stalk cells attenuates the VEGF-induced ERK and p21 activation. In the more mature and quiescent vascular areas, most ECs have very low P-ERK levels and have already exited the cell cycle (KI67−) because they are exposed to less VEGF. However, these cells still have active Notch signalling, which is important for maintaining their quiescence. **c** When Dll4/Notch signalling is inhibited, angiogenic stalk cells display a tip-cell like profile and upregulate Esm1 expression and ERK phosphorylation. This leads to p21 expression, which induces the cell-cycle arrest of stalk-cells, and compromises the subsequent proliferation and development of vessels toward the hypoxic tissue area. Impairment of Dll4/Notch signalling in more mature and quiescent ECs (grey), produces a distinct effect; here, the increase in P-ERK after Dll4/Notch inhibition is more moderate and productive, and ECs enter the cell cycle even when VEGF signalling is low

with the Neural Tissue Dissociation Kit (Mylteny 130-092-628). The resulting cell suspension was incubated with conjugated APC-anti-CD31 antibody (BD Pharmingen, 551262), followed by DAPI staining to exclude dead cells. Viable (DAPI−) ECs (APC-CD31+) were FACS sorted and processed for total RNA extraction with the RNAeasy Micro kit (Qiagen). Between 300 and 2500 ECs were collected per sample.

**RNA isolation and qRT-PCR**. A number of strategies were used to profile gene expression in retinal ECs subjected to different Notch and VEGF signalling modulations in vivo. Firstly, we treated neonatal mouse pups with control IgG or with anti-Dll4 or anti-VEGF antibodies. Secondly, we treated P5 mice with vehicle or DAPT for 24 h. Thirdly, we induced *Rbpj* deletion between P1 and P3 and collected fluorescent ECs by FACS at P6, as described above. Four retinas of at least

two animals were processed independently. In total, 12 retinas (6 animals/3 samples per group) were dissected and 300–2500 CD31-APC+ ECs per sample (with or without other fluorescent protein expression) were sorted for RNA extraction using the RNAeasy Micro Kit.

For quantitative real time PCR (qRT-PCR), RNA extracted from the ECs obtained as above was retrotranscribed with the High Capacity cDNA Reverse Transcription Kit with RNase Inhibitor (Thermo fisher, 4368814). cDNA was preamplified with Taqman PreAmp Master Mix containing a Taqman Assay-based pre-amplification pool containing a mix of the following Taqman assays (Applied Biosystems): *Actb, Gapdh, Pecam1, Cdh5, Hey1, Hey2, EphrinB2, Rbpj, Dll4, Angpt2, Esm1, Apln, Cdk2, Cdk4, Cdkn1a, Cdkn1b, Trp53, Vegfr2/Kdr, Vegfr1/Flt1, and Vegfr3/Flt4*. Preamplified cDNA was used in standard qRT-PCR with gene-specific Taqman Assays (Thermo fisher) in a AB7900 thermocycler (Applied Biosystems).

**In vitro assessment of the role of Notch in EC proliferation**. HUVECs (50.000 cells, Lonza) were plated in a 96-well plate precoated with 0.2% gelatin and cultured for 2 days in Endothelial Cell Growth Medium (PromoCell C22010). For the experiment, the medium was exchanged and supplemented with VEGFA at a final concentration of 50 ng/mL plus 10 μM dibenzazepine (DBZ, YO-01027 Selleckchem) or an equal volume of DMSO (vehicle only control); cells were incubated for an additional 24 h. At 4 h before the end of the experiment (20 h timepoint), EdU (8 μM) was added to the medium. Cells were then washed with PBS and fixed with 4% PFA for 10 min at RT, permeabilized for 10 min with TBST (25 mM Tris HCl pH 7.4, 150 mM NaCl, 0.5% Triton X-100), blocked with TBST containing 2% BSA and 2% donkey serum and incubated at 4 °C overnight with rabbit anti-Ki67 (1:200, Thermo Fisher, RM-9106-S0). The following day, cells were washed three times with TBST, incubated with Cy3-conjugated donkey anti-rabbit and Hoechst for 2 h at RT, washed twice with TBST, stained for EdU using the Click-iT EdU Alexa Fluor 647 kit, washed twice with PBS, and mounted in Fluoromount. Images were taken in a Zeiss LSM 700 confocal microscope.

To generate mouse ECs expressing the FUCCI reporter[58], we first generated G4 mouse embryonic stem (ES) cells with CRISPR/Cas9-assisted knock-in of the FUCCI construct in the Rosa26 locus, using a recently established method[13]. These ES cells were later used to generate embryoid bodies (EBs) following published protocols[13]. EBs were collected and deposited on an OP9 cell monolayer (a stromal cell line from mouse bone marrow —ATCC CRL-2749) and cultured for 5 days in OP9 cell medium (alfa-MEM + 20% FBS). OP9 medium was changed every 2 days. To enhance endothelial differentiation, the OP9 medium was supplemented with human VEGF-A (Peprotech) at a final concentration of 30 ng/μl. At day 5, the Notch inhibitor DBZ was added to the medium at a concentration of 10uM, 24 h before the FACs sorting analysis. To analyse Venus fluorescence in the endothelial population by FACS, cells were trypsinised for 20 min and resuspended for 30 min in blocking buffer (DPBS + 5% liophilized FBS). Cells were then respun and resuspended for 30 min at 4 °C in the same buffer containing rat anti-ICAM2 (#553325, BD Biosciences), followed by incubation for 15 min at 4 °C with Donkey anti-rat Alexa 647 conjugated antibody (Jackson laboratories, 1:200). Finally, cells were washed with blocking buffer to eliminate unbound antibody, and the antibody and Venus fluorescent signals were taken analysed by flow cytometry.

**Microscopy**. We used different laser-scanning confocal microscopes depending on the complexity of the immunostainings and the combination of fluorescent proteins detected. For multicolour detection of up to 7 signals, we used the inverted Leica TCS SP5 confocal (405, 488, 514, 546, 594 and 633 nm) or the Leica TCS SP8 confocal with a 405 nm laser and a white laser that allows excitation at any wavelength from 470 to 670 nm. Confocal ZEISS LSM780 was also used sporadically. For the mouse retina analysis, we always used laser scanning confocal analysis with a 10×, 20×, or 40× lens. We acquired individual fields or tiles of large areas. All images shown are representative of the results obtained for each group and experiment. Littermates were dissected and processed under exactly the same conditions. Comparisons of phenotypes or signal intensity were made with pictures obtained using the same laser excitation and confocal scanner detection settings. Images were processed using ImageJ/Fiji and Adobe Photoshop.

**Quantitative analysis of retinal vasculature**. Single-low magnification (10× lens) or tile merged high magnification (20× or 40× lens) confocal fields of the angiogenic front or the mature vascular area were quantified with Fiji/ImageJ. Each microscopy field contained hundreds of ECs, and the relative or absolute number of cells in each field is indicated in the charts by a dot. As indicated in figure legends, microscopy images from several animals and retinas (2 per animal) were used for the phenotypic comparisons and quantifications. Vascular IsolectinB4+ area and Erg+, or Edu+ cells were quantified semiautomatically using custom Fiji macros. EC density (EC number/mm²) was measured as the number of Erg+ cells relative to the vascularised IsolectinB4+ area in each field. The frequencies of Erg+ cells (ECs) in S-phase (EdU+), in the cell cycle (KI67+), or p21+ were determined as the ratio of double-positive cells to the total number of Erg+ cells per field. In some experiments, these signals were quantified in images including membrane- or chromatin-localised fluorescent proteins, enabling identification and quantification

of proliferation frequency and p21 expression in different subsets of mutant and control cells.

**Statistical analysis**. Two groups of samples with a Gaussian distribution were compared by unpaired two-tailed Student $t$ test. Comparisons among more than two groups were made by ANOVA followed by the Turkey pairwise comparison. Graphs represent mean ± SD as indicated, and differences were considered significant at $p < 0.05$. All calculations were done in Excel and final datapoints analysed and represented with GraphPad Prism. No randomisation or blinding was used, and animals or tissues were selected for analysis based on their genotype, the detected Cre-dependent recombination frequency, and quality of multiplex immunostaining. The sample size was chosen according to the observed statistical variation and published protocols.

**Reporting Summary**. Further information on research design is available in the Nature Research Reporting Summary linked to this article.

## Data availability
All data supporting the findings of this study are available from the corresponding author upon request. This includes raw data such as unprocessed original pictures and independent replicates, which are not displayed in the manuscript, but are included in the data analysis in the form of graphs. The source data underlying all figures numeric or chart data are provided as a Source Data file.

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

## Acknowledgements

We thank Simon Bartlett for English editing; José Luis de La Pompa for advice and input throughout the project and for sharing the *Tie2-Cre* mice, I. Flores and R.H. Adams for the *p21*−/− and *Cdh5(PAC)-CreERT2* mice, respectively; Gonzalo Gancedo-Alonso and Sofia Sanchez for assistance with the mouse colony and genotyping; members of the CNIC Gene Targeting and Transgenesis units for the generation of mouse lines, and Genentech for providing the Dll4-blocking antibody. Research in the R.B. group was supported by the European Research Council (ERC-2014-StG—638028), the Centro Nacional de Investigaciones Cardiovasculares (CNIC), and by the Ministerio de Economia, Industria y Competitividad (MEIC: SAF2013-44329-P, SAF2013-42359-ERC and RYC-2013-13209). The CNIC is supported by the Ministerio de Ciencia, Innovación y Universidades (MCNU) and the Pro CNIC Foundation, and is a Severo Ochoa Center of Excellence (SEV-2015–0505). S. Pontes-Quero and M. Fernandez-Chacon received PhD fellowships from the La Caixa bank. W. Luo received a COFUND CNIC International Postdoctoral fellowship.

## Author contributions

S.P.-Q. and R.B. designed experiments and interpreted the results. S.P.-Q. executed most of the experiments. S.P.-Q., M.F.-C., W.L., F.L., I.G.-G., A.H. and R.B. dissected mouse tissues and performed the FACS, immunostainings, microscopy and phenotypic analysis of the different mouse and cell lines. S.P.-Q. and S.F.R. developed methods and ImageJ scripts for assisted or automatic image quantification and analysis. S.F.R. generated DNA constructs and partially edited the paper. V.C.-G., S.F.R., I.G.-G., M.B. and R.B. generated and analysed the ES and endothelial cell lines. R.B. and S.P.-Q. wrote the paper.

## Additional information

**Competing interests:** The authors declare no competing interests.

