## [Peer Review File · Nature Communications]

Reviewers' comments:

Reviewer #1 (Remarks to the Author):

The work by Pontes-Quero and colleagues revisits the integration and interaction of two key controls of angiogenic sprouting and vascular expansion, the VEGF and Notch signalling pathway, operating to control the rate and balance of endothelial migration and proliferation. The authors are experts in this field and have published seminal work on both VEGF receptor signalling, the various receptors and their importance for sprouting, and on the role and integration of Notch signalling in this process. They are also experts in mouse genetics and in the detailed analysis of the mouse retina as model for angiogenic sprouting.

Previous work established that VEGF quantitatively drives both migration and proliferation of endothelial cells, through the activation of VEGFR2, and that Notch signalling through Dll4 critically regulates which cells responds how, as high levels of Notch activity inhibit the migratory tip cell phenotype. Inhibition of Notch signalling has repeatedly been shown to result in dramatic hypervascularity associated with a dramatic increase in tip cell numbers and associated filopodia and cell migration. In vitro studies also suggested that the lack of Notch activity leads to increased cell numbers as Notch activity inhibited endothelial proliferation. Short term EDU experiments in vivo confirmed this idea. Nevertheless, whereas the tip cells have low Notch activity, they reportedly proliferate less than the highly Notch active stalk cells, seemingly incompatible with the idea that Notch limits endothelial proliferation. The finding that the tip and stalk cell phenotypes are transient and interchanging provided some reconciliation to this conundrum, as it could be assumed that Notch levels will dynamically fluctuate and thus enable cell cycle entry of stalk cells in a low phase. However, the true mechanisms regulating tip and stalk cell proliferation in vivo and the integrative balance of the underlying processes have not been resolved.

Pontes-Quero and colleagues now used novel sophisticated genetic models based on multilox recombination events that drive either high or low levels of Notch or VEGFR2 activation using overexpression or dominant negative approaches combined with cell tracking possibility. This methodology allows for unprecedented control and insight into cell-autonomous effects and the resulting expansion of clones or collectives of cells in the developing retinal vasculature. Using inducible Cre drivers that either activate in all endothelial cells or just in the ESM1 positive tip cells, and tamoxifen pulsing to recombine many or just a few cells, the authors interrogated whether and how the different levels of Notch and VEGFR2 activity alter the numbers of cells, clone sizes, cell cycle entry or exit and thus gain detailed insight into the dynamics and duration of cell cycle. The ability to track the cells also enabled the authors to isolate cells by FACS sorting, and further study gene expression levels to identify potential regulators of cell cycle.

The authors identified, contrary to common assumption, that reducing Notch activity in cells at the sprouting front reduces cell proliferation, as cells accelerate their cycle and then rapidly exit into cell cycle arrest. Inhibiting Notch in the more established vasculature that is under lower VEGF stimulation however leads to increased cell proliferation and not to cell cycle arrest. The authors find that this is caused by VEGF driven levels of p-ERK activity, which in turn drives expression of p21. The authors rigorously test this idea and provide convincing evidence for their model, showing that driving or reducing VEGFR2 activity cell autonomously shows the same bimodal effects. Intriguingly, seemingly as biproduct of their work, the authors confirm many of the fundamental concepts providing solid quantitative data on tip and stalk cell proliferation ratios, but now demonstrate the molecular mechanism why tip cells normally proliferate less than stalk cells although they show highest activation levels by the mitogenic stimulus of VEGF.

The work is of outstanding quality and of fundamental importance for the deeper understanding of the key drivers of angiogenesis. The quantitative biology and the identification of the differential effects of Notch activity depending on the level of VEGF activation solves a number of long standing questions in the field, and the new insights into p21 and ERK activity in this scenario will likely open up better possibilities to modulate angiogenesis in therapy.

Overall, the work is extremely well written, and the sometimes complicated technological aspects are well explained and illustrated. Methods and statistics are well described. This is probably the first time I recommend a manuscript to be published without delay and with no revisions.

This work will be influential beyond the immediate specialist field in angiogenesis as it provides insights into fundamental quantitative biology and expands the tool kit for in vivo quantitative biology work tremendously.

Reviewer #2 (Remarks to the Author):

The authors add another very interesting piece to our current understanding of tip and stalk cell selection during angiogenesis. By using a very elegant genetic reporter system, which was recently published in *Cell* by the same group allowed to trace individual clones of endothelial cells within the postnatal mouse retina, which represents the best model system to study angiogenesis in rodents. The authors perform gene manipulation experiments to interfere with the most central regulators of angiogenesis (VEGF and DLL4/Notch/Rbpj). Very surprisingly, they find that high mitogenic stimuli rather arrest angiogenesis due to Notch-driven p21 expression leading to cell cycle arrest. This is pretty astonishing but the experiments seem to be well executed and well controlled. Data presentation is excellent and the manuscript is well written.

The main weakness of this work is the use of rather artificial non-physiological gene dosing manipulation. It is unlikely that such uniform Notch gain or loss of function would ever occur in a natural setting. The outcome of Notch signaling is strictly context-dependent and dosis-dependent. There is even evidence about oscillating Notch target gene expression.

Major points:

1) The authors should clearly state in the discussion of this paper that their experimental approach is somehow artificial. Alternatively, they should proof that such a scenario really occurs in a physiological setting or in a disease model.

2) In Figure 1C there is some indication that there are less EC at the angiogenic front after KD of Rbpj. However, blocking Dll4 (what also inhibits Notch signaling) leads to the opposite effect in Fig 2E. This requires a much deeper analysis to clarify the contradictory results. Can we authors also block the main Notch receptor (Notch1) to shed more light on this?

3) In Figure 2 the authors show that there is a transient increase in EC numbers between 24h and 48 post anti-DLL4 treatment and this increase does not correspond to an increase in frequency of proliferating ECs. They propose that this could be due to higher speed of the endothelial cell cycle. However, data to proof this theory are missing. This could be further addressed by performing cell culture studies including live cell imaging.

Another idea would be to isolate primary EC from the mosaic reporter mice and perform 2D or even 3D cultures using VEGF stimulation, Notch activation or blocking and cell cycle as well as live cell microscopy as read-out.

4) There are major inconsistencies in the EdU staining between panels E and F in Figure 3.

5) In Figure 4 the authors conclude from panels C and D that tip cells eventhough they are exposed to more VEGF protein exit the cell cycle and produce fewer clones. On the contrary, stalk cells proliferate more and produce larger clones. However, how do the authors rule out the possibility the these "clones" are derived from a single cell and not the result of fusing two adjacent clones labeled with the same color?

6) From Figure 5 E the authors conclude that p21 upregulation followed by Notch LoF is cell autonomous. However, there are multiple Tomato-positive cells which are at the tip position but

have low p21 expression. This requires a much better explanation or one should better tone-down the statement about a strict cell autonomous mechanism.

Minor:

In Figure 6 the scale bars are missing.

Reviewer #3 (Remarks to the Author):

This paper describes how the balance of VEGF and Notch signaling controls the proliferation of endothelial cells during angiogenesis. Previous work has revealed an intriguing paradox whereby tip cells are exposed to the highest levels of VEGF yet may not proliferate as much as stalk cells, which are exposed to lower levels of VEGF. In this paper, the authors use several mouse models to induce various levels of both VEGF and Notch signaling in single cells during angiogenesis and look at cell proliferation. The authors find that moderate VEGF signaling induces sustained proliferation and quite unexpectedly, high VEGF signaling induces cell cycle arrest. The authors conclude that there is a bell-shaped dose response to VEGF signaling, and at high doses, VEGF and ERK induce p21 expression and cell cycle exit.

I find the mouse models and the techniques used in this paper to be quite powerful and innovative. I think the ability to generate in a single mouse, cells that have different levels of Notch and VEGF signaling to be a perfect way to test the authors stated hypothesis. I also felt that being able to look at single-clones within the retina and how they proliferate is quite informative. I think the tools generated for this study would also be useful to the broader scientific community and cell cycle community since it allows for the in vivo study of cell proliferation.

My main concern is the conclusion that high VEGF and ERK signaling leads to an upregulation of p21 and cell cycle exit. Since ERK signaling is not thought to directly lead to p21 upregulation, the observed increase in p21 expression is likely through an indirect effect. The authors report that inhibition of Notch signaling leads to a short burst in proliferation over 24 hours, but then the cells have exited the cell cycle by 48 hours. So its likely that the cells are able to go through 1 cell cycle but then the resulting daughter cells exit to quiescence. This might indicate that the cells could be damaged or stressed out, and that is why they are no longer proliferating. The authors should check for the presence of DNA damage (gammaH2AX staining or 53BP1 foci) or p53 activation, which is known to directly transcriptionally induce p21. Consistent with this hypothesis, the authors state they think the cell cycle is ultrafast when they artificially inhibit Notch signaling. So perhaps this fast

cell cycle leads to DNA damage and ultimately cell cycle exit. The authors could also block p53 and see if that is able to rescue the cell cycle exit phenotype.

Minor points: The manuscript could benefit from a few extra small schematics for the mosaic experiments to make them easier to understand. For example, Fig 1 F could benefit from a table or a scheme showing Red cells have normal Notch signaling, Green cells have decreased Notch signaling, and Cyan cells have increased Notch signaling. This was a really nice experiment but it took me awhile to understand which cells where which.

Answers to Reviewers NCOMM

Reviewers' comments:

Reviewer #1 (Remarks to the Author):

The work by Pontes-Quero and colleagues revisits the integration and interaction of two key controls of angiogenic sprouting and vascular expansion, the VEGF and Notch signalling pathway, operating to control the rate and balance of endothelial migration and proliferation. The authors are experts in this field and have published seminal work on both VEGF receptor signalling, the various receptors and their importance for sprouting, and on the role and integration of Notch signalling in this process. They are also experts in mouse genetics and in the detailed analysis of the mouse retina as model for angiogenic sprouting.

Previous work established that VEGF quantitatively drives both migration and proliferation of endothelial cells, through the activation of VEGFR2, and that Notch signalling through Dll4 critically regulates which cells responds how, as high levels of Notch activity inhibit the migratory tip cell phenotype. Inhibition of Notch signalling has repeatedly been shown to result in dramatic hypervascularity associated with a dramatic increase in tip cell numbers and associated filopodia and cell migration. In vitro studies also suggested that the lack of Notch activity leads to increased cell numbers as Notch activity inhibited endothelial proliferation. Short term EDU experiments in vivo confirmed this idea. Nevertheless, whereas the tip cells have low Notch activity, they reportedly proliferate less than the highly Notch active stalk cells, seemingly incompatible with the idea that Notch limits endothelial proliferation. The finding that the tip and stalk cell phenotypes are transient and interchanging provided some reconciliation to this conundrum, as it could be assumed that Notch levels will dynamically fluctuate and thus enable cell cycle entry of stalk cells in a low phase. However, the true mechanisms regulating tip and stalk cell proliferation in vivo and the integrative balance of the underlying processes have not been resolved.

Pontes-Quero and colleagues now used novel sophisticated genetic models based on multilox recombination events that drive either high or low levels of Notch or VEGFR2 activation using overexpression or dominant negative approaches combined with cell tracking possibility. This methodology allows for unprecedented control and insight into cell-autonomous effects and the resulting expansion of clones or collectives of cells in the developing retinal vasculature. Using inducible Cre drivers that either activate in all endothelial cells or just in the ESM1 positive tip cells, and tamoxifen pulsing to recombine many or just a few cells, the authors interrogated whether and how the different levels of Notch and VEGFR2 activity alter the numbers of cells, clone sizes, cell cycle entry or exit and thus gain detailed insight into the dynamics and duration of cell cycle. The ability to track the cells also enabled the authors to isolate cells by FACS sorting, and further study gene expression levels to identify potential regulators of cell cycle.

The authors identified, contrary to common assumption, that reducing Notch activity in cells at the sprouting front reduces cell proliferation, as cells accelerate their cycle and then rapidly exit into cell cycle arrest. Inhibiting Notch in the more established vasculature that is under lower VEGF stimulation however leads to increased cell proliferation and not to cell cycle arrest. The authors find that this is caused by VEGF driven levels of p-ERK activity, which in turn drives expression of p21. The authors rigorously test this idea and provide convincing evidence for their model, showing that driving or reducing VEGFR2 activity cell autonomously shows the same bimodal effects. Intriguingly, seemingly as byproduct of their work, the authors confirm many of the fundamental concepts providing solid quantitative data on tip and stalk cell proliferation ratios, but now demonstrate the

molecular mechanism why tip cells normally proliferate less than stalk cells although they show highest activation levels by the mitogenic stimulus of VEGF.

The work is of outstanding quality and of fundamental importance for the deeper understanding of the key drivers of angiogenesis. The quantitative biology and the identification of the differential effects of Notch activity depending on the level of VEGF activation solves a number of long standing questions in the field, and the new insights into p21 and ERK activity in this scenario will likely open up better possibilities to modulate angiogenesis in therapy.

Overall, the work is extremely well written, and the sometimes complicated technological aspects are well explained and illustrated. Methods and statistics are well described. This is probably the first time I recommend a manuscript to be published without delay and with no revisions.

This work will be influential beyond the immediate specialist field in angiogenesis as it provides insights into fundamental quantitative biology and expands the tool kit for in vivo quantitative biology work tremendously.

A: We thank the reviewer for such a fantastic and detailed summary of our work. And the very insightful comments as well. These are the words that motivate young scientists and keep them moving forward.

Reviewer #2 (Remarks to the Author):

The authors add another very interesting piece to our current understanding of tip and stalk cell selection during angiogenesis. By using a very elegant genetic reporter system, which was recently published in Cell by the same group allowed to trace individual clones of endothelial cells within the postnatal mouse retina, which represents the best model system to study angiogenesis in rodents. The authors perform gene manipulation experiments to interfere with the most central regulators of angiogenesis (VEGF and DLL4/Notch/Rbpj). Very surprisingly, they find that high mitogenic stimuli rather arrest angiogenesis due to Notch-driven p21 expression leading to cell cycle arrest. This is pretty astonishing but the experiments seem to be well executed and well controlled. Data presentation is excellent and the manuscript is well written.

The main weakness of this work is the use of rather artificial non-physiological gene dosing manipulation. It is unlikely that such uniform Notch gain or loss of function would ever occur in a natural setting. The outcome of Notch signaling is strictly context-dependent and dose-dependent. There is even evidence about oscillating Notch target gene expression.

Major points:

- 1) The authors should clearly state in the discussion of this paper that their experimental approach is somehow artificial. Alternatively, they should prove that such a scenario really occurs in a physiological setting or in a disease model.

A: We thank the reviewer for appreciating the high quality of the work and data presentation. We may not have fully understood the comment on the artificial experimental approach and apologize in advance if our words herein do not fully address the concern raised. We believe our experimental approaches are as artificial as any other genetic gain or loss-of-function approach. Perhaps the reviewer is referring to the use of mosaic dominant-negative and gain-of-function approaches (iChrNotch and iMb-Vegfr2-Mosaics). However, the results obtained with these new mouse models,

were validated with well established/published mouse lines (i.e. *Dll4^{flox}* and *Rbpj^{flox}*) and pharmacological compounds (Anti-Dll4). Thus, in addition to the mosaic and single-cell approaches, we also used classical and uniform gene gain and loss-of-function approaches.

We fully agree that in a natural setting, the uniform Notch gain or loss-of-function does not occur, as with most other signalling pathways. There are also dozens of papers describing the importance of Notch signalling oscillations and dose for cell biology. However, uniform loss-of-Notch signalling does occur when pharmacological compounds (i.e. γ -secretase inhibitors or anti-Dll4) are used to target Notch signalling therapeutically.

Importantly, we also characterized and validated the same mechanisms in cells with physiological mitogenic/P-ERK and VEGF/Notch signalling levels, such as stalk-cells and *Esm1+* tip cells (Fig. 4 and Fig. 7) of wild-type retinas.

Our work provides important and unexpected new insights on the molecular and cellular outcome of inhibiting Notch signalling, or activating VEGF signalling, in therapeutic settings.

2) In Figure 1C there is some indication that there are less EC at the angiogenic front after KD of *Rbpj*. However, blocking Dll4 (what also inhibits Notch signaling) leads to the opposite effect in Fig 2E. This requires a much deeper analysis to clarify the contradictory results. Can we authors also block the main Notch receptor (Notch1) to shed more light on this?

A: We agree that on a first look the results may seem contradictory. However, they are not, and we were careful to include extra data in Sup. Fig. 3, and write a few sentences in the text related with that (please read manuscript quote further below).

The difference between 1C and 2E is due to the different temporal resolution in the phenotypic analysis. In Figure 1C, we analyse the effect of deleting *Rbpj* in endothelial cells from P1 to P6 (5 days). In Figure 2E we analyse the effect of blocking Dll4/Notch signalling for 24h and 48h. The *Rbpj* LOF analysis is a longer-term loss-of-function analysis. Our results show that the cell-cycle arrest after Dll4/Notch inhibition occurs 24h-to-48h after a transient period of increased proliferation, which explains the difference between the short (Fig. 2E) and longer-term (Fig. 1C) phenotypes. In accordance with this, we show in Sup. Fig. 3 that treatment of pups with anti-Dll4 for 72h, results in less endothelial density at the angiogenic front, similarly to deletion of *Rbpj* from P1 to P6. Therefore, the results are not contradictory. Below is the paragraph from the manuscript that is related with this data.

“Importantly, this burst in vascular expansion was transient, lasting less than 48 hours, by which time most ECs with loss of Notch signalling have already exited the cell cycle (Fig. 2d, f and Supplementary Fig. 4d, f). In contrast, in vessels of control IgG treated animals, the slower-dividing ECs, with normal Notch levels, continued to cycle for longer (Fig. 2c, f and Supplementary Fig. 4d, 4f), generating more ECs overtime (more ERG+ ECs per IsolectinB4 area) when compared with retinas treated with the Dll4 blocking antibody for 72h (Supplementary Fig. 3a, b), mimicking the effect of *Rbpj* deletion from P1 to P6 (Fig. 1a, b)”.

Regarding the suggestion to perform new experiments with Notch1 floxed mice: We already show Dll4/Notch signalling loss-of-function data with Anti-Dll4, DN-Maml1, Dll4 full genetic deletion and *Rbpj* full genetic deletion. Besides this, angiogenic retina endothelial cells express four Notch receptors with the following CPM (count per million): Notch1 (430 CPM), Notch2 (80 CPM), Notch3

(200 CPM) and Notch4 (185 CPM). Therefore, it is likely that the retina Notch1 LOF phenotype is milder and distinct from Dll4 or Rbpj deletion phenotypes. There is also published evidence of this in Benedito et al., 2012 Nature.

3) In Figure 2 the authors show that there is a transient increase in EC numbers between 24h and 48h post anti-DLL4 treatment and this increase does not correspond to an increase in frequency of proliferating ECs. They propose that this could be due to higher speed of the endothelial cell cycle. However, data to proof this theory are missing. This could be further addressed by performing cell culture studies including live cell imaging.

Another idea would be to isolate primary EC from the mosaic reporter mice and perform 2D or even 3D cultures using VEGF stimulation, Notch activation or blocking and cell cycle as well as live cell microscopy as read-out.

A: We show that there is an increase in the number of endothelial cells between 24 and 48h post antiDLL4 treatment, and no difference in the frequency of cells in s-phase (EdU+) or in cycle (Ki67+) at 12h (Sup. Fig. 3 G-H) and 24h (Fig. 2) timepoints. We (and others) have also performed the standard apoptosis assays, and never found any apoptosis in endothelial cells during retina vascular development. Given these results, if after anti-Dll4 the number of cells increase in the 24h-48h timepoints, without a change in the frequency of proliferating cells or apoptosis, we do not conceive any other hypothesis than that there is an increase in the average speed of endothelial proliferation. Our data also indicates that most of the increase in EC number occurs in the first 24h (Fig. 2), which is consistent with the upregulation of p21 and cell cycle exit observed between the 24h-48h timepoints. Nonetheless, in the manuscript text related with these results, we were careful to use the word “suggest” instead of show or indicate. Live imaging of zebrafish embryos by the Nathan Lawson lab also showed increased EC proliferation in ISVs of Dll4 morpholino treated embryos (Siekmann and Lawson, 2007).

In the last years, we (and others) have done numerous *in vitro* experiments with Notch inhibitors. Some of those results obtained in our laboratory are in Sup. Fig.1 or published in Pontes-Quero et al., 2017 Cell. *In vitro*, we see that Notch inhibition (Sup. Fig. 1), or expression of DN-Maml1, increases EC proliferation (Figure 2E in Pontes-Quero et al., 2017 Cell). However, the cell-cycle arrest mechanism induced by Notch inhibition or VEGF overactivation identified here *in vivo*, does not occur *in vitro*. Several labs have also shown that ECs cultured with high and saturating levels of VEGF ligands do not arrest in cell-cycle, they just reach a maximum/plateau level of proliferation.

This difference between the *in vitro* and *in vivo* findings was already discussed in our manuscript: “Contrasting our *in vivo* data, Notch signalling inhibition, or increased VEGF signalling in endothelial cell lines always increases their proliferation (Sup. Fig. 1)^{22, 23}, suggesting that the bell-shaped doseresponse and cell-cycle arrest mechanism we identified here is inactive *in vitro*, or is bypassed. This discrepancy may reflect significant differences between the *in vitro* and *in vivo* biochemical and physical endothelial microenvironment, such as shear stress, oxygen concentration, EC-to-EC junctional stability and nutrient availability. All these factors might interact with and modify the operation of the cell cycle arrest mechanism identified”.

Interestingly, the results obtained by us and others with Notch inhibitors *in vitro*, seem to match more the effect of Notch inhibition in mature/quiescent ECs *in vivo*, in which it induces EC proliferation. It is also important to note, that when HUVECs or MECs have some Notch signalling *in vitro* (never as much as vessels *in vivo*), they are highly confluent in order to enable cell-to-cell Notch signalling. In this context of *in vitro* cell confluency, we see that only around 10% of HUVECs or MECs are Ki67+ . This frequency increases after Notch LOF to 25% (HUVECs) and 46% (MECs) as shown in Sup. Fig. 1. However *in vivo*, 96% of angiogenic stalk-cells are Ki67+ (Sup. Fig. 4F) and after 48h of

Dll4/Notch LOF that frequency decreases to 9%. This data suggests that ECs growing *in vitro*, with regular serum, do not display the same mitogen/ERK induced p21 mediated checkpoint and cell-cycle dynamics.

Our data also shows that the cell-cycle arrest process we identified *in vivo*, is not dependent on cell density, or collective loss or gain of-signalling, since single cells with loss of Rbpj, or increase in VEGFR2 signalling, upregulate p21 and exit cell-cycle, even when surrounded by wildtype cells (Fig.5E-5H and Fig. 3C-3I and Fig. 4E-4G).

The sum of this data make us believe that the *in vivo* endothelial cell cycle dynamics of growing angiogenic vessels is significantly different from what can be modelled *in vitro*. If the cell cycle arrest mechanism we found, occurred *in vitro*, we believe it would have been published already several years ago.

We understand the limitations of our *in vivo* mouse genetic experiments and the lack of mouse live imaging to really get the *in vivo* cell-cycle dynamics of the process, but in this particular case, we think that resorting to *in vitro* analysis (most of which we did already) will not allow us to better understand what occurs during angiogenesis *in vivo*.

4) There are major inconsistencies in the EdU staining between panels E and F in Figure 3.

A: The panels E and F have pictures illustrating at high resolution/magnification the different signals detected. Quantifications were performed on many more pictures of larger size.

E and F show ERG and EdU stainings with distinct color combinations. The pink may be more difficult to appreciate in a 3-channel picture than the yellow. However, quantifications were done with Image J/FIJI and using grey signal channels.

5) In Figure 4 the authors conclude from panels C and D that tip cells eventhough they are exposed to more VEGF protein exit the cell cycle and produce fewer clones. On the contrary, stalk cells proliferate more and produce larger clones. However, how do the authors rule out the possibility the these “clones” are derived from a single cell and not the result of fusing two adjacent clones labeled with the same color?

A: We believe the reviewer’s question arises from looking at the high magnification pictures provided in Fig. 4C, particularly the IgG control figure. We wanted to show an example with high resolution and enough detail of 2 independent clones having a different number of cells, for the reader to have more visual representation of the variability in the wildtype single-cell clonal expansion, but we fully understand the reviewer comment and that the selected figure can be wrongly interpreted. When we prepared the figure panels, we had exactly the same concern as the reviewer, and that is why we also included in Fig. 4C left, one low-magnification picture, having one single clone of 2 cells. As explained in the manuscript text, recombination with our Esm1-H2BCerulean-2A-iCreERT2 line results “... in very few recombined clones per retina arising from a few Esm1+ tip cells (Fig. 4C left), enabling us to score and assign single-tip-cell-derived clones over a 4day period”.

To highlight this better, we now provide four more low magnification pictures in Sup. Fig. 7B, where it is more clear that our quantifications were done in retinas with very few recombined tip cells. Given the very low frequency of Tomato+ cells in the entire tissue, and their clonal and nearby distribution, it is statistically highly improbable that clones with more than 2 cells arised from more than one independent stochastic recombination event in more than one Esm1+ tip cell. In addition to the new Sup.Fig. 7B, we now include in Fig. 4C new representative figures showing only one (not two) single-cell derived clone.

We also decided to incorporate more data and better figures in Fig. 4d and Sup. Fig. 7c. Our previous data was obtained with the 1st generation Dual ifg mosaic mice, and only 30 clones were quantified at that time. We have included now data obtained with the 2nd generation Dual ifgMosaic mice (Pontes-Quero et al., 2017) that represent the quantification 169 single-cell derived stalk cell clones at the angiogenic front. We briefly explain the published method in the text and in the new figure 4d and Sup. Fig. 7c legends. More details about the method can be seen in Pontes-Quero et al., 2017, Cell.

6) From Figure 5 E the authors conclude that p21 upregulation followed by Notch LoF is cell autonomous. However, there are multiple Tomato-positive cells which are at the tip position but have low p21 expression. This requires a much better explanation or one should better tone-down the statement about a strict cell autonomous mechanism.

A: Indeed not all single MbTomato/RbpjKO cells are p21+ in a given temporal snapshot (P6), even though they all lost Rbpj expression (Sup. Fig. 1E). Only 48% of the angiogenic front (AF) or 85% of leading edge (LE) Tomato+/RbpjKO cells are p21+ (Fig. 5F). However, these numbers contrast dramatically with the % of wildtype cells that are p21+, 8% AF and 21% LE (Fig. 5G).

This data either suggests that a fraction of Rbpj LOF cells loose p21 expression after the cell cycle arrest, or that there is single-cell heterogeneity as suggested by the reviewer. An alternative hypothesis, which we favour, is that not all Tomato+/RbpjKO cells at the angiogenic front or leading edge, will have high VEGF/ERK signalling. Indeed, this is clearly demonstrated by our analysis of the Esm1 reporter (Fig. 4) and P-ERK (Fig. 5). Accordingly, in the more mature vascular area, where VEGF levels are significantly lower, only 6% of MbTomato-RbpjKO cells have p21 expression (Fig. 5F). We still think that our data strongly suggests a single-cell autonomous, cell-cycle arrest mechanism. Unfortunately, the pulse-and-chase nature of the Rbpj LOF *in vivo* analysis, does not allow us to know if all angiogenic MbTomato+/Rbpj KO cells expressed p21 at some point between P1 and P6, we only see the endpoint (P6). But the difference between single mutant (Fig. 5F) and wildtype (Fig. 5G) cells is very significant. (please note that in Rbpj flox mutants we cannot assure that all MbTomato- cells are wildtype, a small fraction may have Rbpj deletion, and this explains the difference in the white bars between Fig. 5F and 5G).

Following the reviewer advice, we inserted the following paragraph in the relevant section:

“Interestingly, only a fraction of single angiogenic ECs with full Dll4/Notch or Rbpj LOF had high p21 levels at any given timepoint of analysis (Fig. 4i, 4j Anti-Dll4 (24h) and Fig. 5f Rbpj deletion (P1-P6)), suggesting single-cell heterogeneity in p21 upregulation. This heterogeneity, may be related with the known heterogenous and highly dynamic response of ECs to VEGF signalling⁴⁵, and is also evident in the Esm1 reporter and P-ERK immunostaining analysis presented above (Fig. 4a, 4b and Fig. 5b).”

Minor:

In Figure 6 the scale bars are missing.

A: We included them in the revised manuscript.

Reviewer #3 (Remarks to the Author):

This paper describes how the balance of VEGF and Notch signaling controls the proliferation of endothelial cells during angiogenesis. Previous work has revealed an intriguing paradox whereby tip

cells are exposed to the highest levels of VEGF yet may not proliferate as much as stalk cells, which are exposed to lower levels of VEGF. In this paper, the authors use several mouse models to induce various levels of both VEGF and Notch signaling in single cells during angiogenesis and look at cell proliferation. The authors find that moderate VEGF signaling induces sustained proliferation and quite unexpectedly, high VEGF signaling induces cell cycle arrest. The authors conclude that there is a bell-shaped dose response to VEGF signaling, and at high doses, VEGF and ERK induce p21 expression and cell cycle exit.

I find the mouse models and the techniques used in this paper to be quite powerful and innovative. I think the ability to generate in a single mouse, cells that have different levels of Notch and VEGF signaling to be a perfect way to test the authors stated hypothesis. I also felt that being able to look at single-clones within the retina and how they proliferate is quite informative. I think the tools generated for this study would also be useful to the broader scientific community and cell cycle community since it allows for the *in vivo* study of cell proliferation.

A: We thank the reviewer for appreciating the high quality of the work and the broad relevance of the new genetic tools used. The mouse lines took a long time to generate and also required a very careful quantitative analysis of multiple single-cell, and mosaic, loss and gain-of-function phenotypes *in vivo*. We also used other well established/published mouse lines (i.e. $Dll4^{flox}$ $Rbpj^{flox}$) and pharmacological compounds (Anti-Dll4), to backup the data obtained with the new mouse lines.

My main concern is the conclusion that high VEGF and ERK signaling leads to an upregulation of p21 and cell cycle exit. Since ERK signaling is not thought to directly lead to p21 upregulation, the observed increase in p21 expression is likely through an indirect effect. The authors report that inhibition of Notch signaling leads to a short burst in proliferation over 24 hours, but then the cells have exited the cell cycle by 48 hours. So its likely that the cells are able to go through 1 cell cycle but then the resulting daughter cells exit to quiescence. This might indicate that the cells could be damaged or stressed out, and that is why they are no longer proliferating. The authors should check for the presence of DNA damage (gammaH2AX staining or 53BP1 foci) or p53 activation, which is known to directly transcriptionally induce p21. Consistent with this hypothesis, the authors state they think the cell cycle is ultrafast when they artificially inhibit Notch signaling. So perhaps this fast cell cycle leads to DNA damage and ultimately cell cycle exit. The authors could also block p53 and see if that is able to rescue the cell cycle exit phenotype.

A: We had exactly the same questions and hypothesis ourselves. This indeed is part of our ongoing research and future directions of a follow-up research project. We have done gammaH2AX and 53BP1 staining in whole mouse retinas before, and we could not detect any clear specific signals for these markers. We also do not know of any publication showing staining for these markers in developing vessels. In contrast to the retina angiogenic vessels, we could see signals for these DNA damage markers in sections of tumor vessels (redacted). However, besides the difference in the immunostaining technique (whole retina vs tumor section immunostaining), in the tumor experiments, Dll4 deletion is carried for 2 weeks, in a context of high VEGF signalling and the tumor microenvironment, which is significantly different.

In the retina, and as the reviewer accurately highlighted, the cell-cycle arrest occurs between 24h to 48h after the loss of Notch signalling. The p21 upregulation is seen already at 24h after the start of Notch inhibition (Fig. 4H-4J). Given this data, our current hypothesis is that in the retina, the p21 upregulation and cell-cycle exit response is too fast to be the direct consequence of accumulation of

DNA damage. We have an alternative hypothesis. We think that endothelial cells, due to their direct contact with oxygen-rich blood, have specific and more sensitive mechanisms to immediately arrest their proliferation, when mitogenic/ERK stimuli is high. This arrest likely prevents DNA and cellular damage by reactive oxygen species, and at the same time also enhances endothelial sprouting. However, the hypothesis raised by the reviewer is also a possibility, and we are trying to refine the techniques used to detect DNA damage in the retina, to see if in fact retina tip cells, or cells with a transient loss of Notch signalling, have more DNA damage. With the advance in DNA sequencing techniques, such as targeted telomeres sequencing, or DNA damage hotspots targeted sequencing, it may be soon possible to detect endothelial DNA damage at higher resolution. We are aware that other groups in the field have interesting data on tumor endothelial DNA damage and chromosomal abnormalities. Those may arise when the checkpoint mechanisms we identified are perturbed and the mitogenic stimuli is high and persistent, which may occur more frequently in the tumor endothelium.

Regarding the analysis of the p53 role, or the dependence of the observed cell cycle arrest mechanism on the p53 function, we have the following comments and data:

1. It is true that most literature places p53 upstream of p21 in the DNA damage response, however several groups have shown that in several cell types and contexts, p21 acts as a master effector of multiple other pathways for promoting anti-proliferative activities that are independent of the classical p53 tumour suppressor pathway (Abbas and Dutta, 2009). One such pathway is the HRAS-Raf-Mapk pathway. The p53-independent transactivation of p21 by activated Ras requires the transcription factor E2F1 (Gartel et al., 2000). E2F1 and E2F3 strongly activate p21 transcription by binding to cis-acting elements between -119 to +16 of the p21 promoter (Gartel et al., 1998; Hiyama et al., 1998). Raf, a downstream effector of Ras, also transactivates p21 independently of p53 (Woods et al., 1997).
2. Our qRT-PCR analysis of retina vessels shows that unlike p21, p53 RNA is not upregulated in the retina endothelium after Dll4/Notch LOF (Sup. Fig. 6B).
3. (redacted)

This data indicates that unlike p21, p53 expression is not regulated by Notch or ERK signalling, and is also not required for the induction of p21 expression in cells with high mitogenic stimuli (tip cells). Therefore, p53 does not seem to be associated with the identified mechanism. It may play a role when cells are subjected to replicative stress for longer periods of time. This will be certainly interesting to address in more detail in another future research project.

Minor points: The manuscript could benefit from a few extra small schematics for the mosaic experiments to make them easier to understand. For example, Fig 1 F could benefit from a table or a scheme showing Red cells have normal Notch signaling, Green cells have decreased Notch signaling, and Cyan cells have increased Notch signaling. This was a really nice experiment but it took me awhile to understand which cells where which.

A: We fully understand the reviewer comment, and have inserted labels indicating the normal, low and high levels in Fig. 1F.

References:

- Abbas, T., and Dutta, A. (2009). p21 in cancer: intricate networks and multiple activities. *Nat Rev Cancer* *9*, 400-414.
- Gartel, A.L., Goufman, E., Tevosian, S.G., Shih, H., Yee, A.S., and Tyner, A.L. (1998). Activation and repression of p21(WAF1/CIP1) transcription by RB binding proteins. *Oncogene* *17*, 3463-3469.
- Gartel, A.L., Najmabadi, F., Goufman, E., and Tyner, A.L. (2000). A role for E2F1 in Ras activation of p21(WAF1/CIP1) transcription. *Oncogene* *19*, 961-964.
- Hiyama, H., Iavarone, A., and Reeves, S.A. (1998). Regulation of the cdk inhibitor p21 gene during cell cycle progression is under the control of the transcription factor E2F. *Oncogene* *16*, 1513-1523.
- Jakobsson, L., Franco, C.A., Bentley, K., Collins, R.T., Ponsioen, B., Aspalter, I.M., Rosewell, I., Busse, M., Thurston, G., Medvinsky, A., *et al.* (2010). Endothelial cells dynamically compete for the tip cell position during angiogenic sprouting. *Nat Cell Biol* *12*, 943-953.
- Siekman, A.F., and Lawson, N.D. (2007). Notch signalling limits angiogenic cell behaviour in developing zebrafish arteries. *Nature* *445*, 781-784.
- Woods, D., Parry, D., Cherwinski, H., Bosch, E., Lees, E., and McMahon, M. (1997). Raf-induced proliferation or cell cycle arrest is determined by the level of Raf activity with arrest mediated by p21Cip1. *Mol Cell Biol* *17*, 5598-5611.

REVIEWERS' COMMENTS:

Reviewer #2 (Remarks to the Author):

The authors have nicely addressed all of my concerns.

Reviewer #3 (Remarks to the Author):

The authors have mostly addressed my concerns about DNA damage or p53 being involved in the upregulation of p21. However, they should include in their Discussion the possibility that DNA damage may be involved but that they have not tested that yet.

REVIEWERS' COMMENTS:

Reviewer #2 (Remarks to the Author): The authors have nicely addressed all of my concerns.

Reviewer #3 (Remarks to the Author): The authors have mostly addressed my concerns about DNA damage or p53 being involved in the upregulation of p21. However, they should include in their Discussion the possibility that DNA damage may be involved but that they have not tested that yet.

A: We included the following in discussion:

The expression of p21 was shown before to be triggered by many different pathways, including DNA-damage and p53, labeling cells undergoing senescence⁵². However, p53 expression did not change after Dll4/Notch inhibition (Supplementary Fig. 6). Even though we cannot entirely rule out a potential contribution of p53 and DNA damage for the observed upregulation of p21, our work shows that p21 expression can be instead directly activated by ERK signalling in ECs (Figure 6), similarly to what was shown before in other cell types^{53, 54}.